# Unconscious categorization of sub-millisecond complex images

**Arnaud Beauny** [1,2,3]*, **Adélaïde de Heering** [1,2,3☯], **Santiago Muñoz Moldes** [1,2,3☯], **Jean-Rémy Martin** [1,2,3☯], **Albert de Beir** [4], **Axel Cleeremans** [1,2,3]

**1** Consciousness, Cognition & Computation Group (CO3), Université Libre de Bruxelles (ULB), Brussels, Belgium, **2** Center for Research in Cognition & Neurosciences (CRCN), Université Libre de Bruxelles (ULB), Brussels, Belgium, **3** ULB Neuroscience Institute (UNI), Université Libre de Bruxelles (ULB), Brussels, Belgium, **4** Robotics and Multibody Mechanics Research Group, Vrije Universiteit Brussel and Flanders Make, Brussels, Belgium

☯ These authors contributed equally to this work.
* arnaud.beauny@gmail.com

**Data Availability Statement:** All data used in the manuscript and supplementary material are available from the database following this Open Science Framework link: https://osf.io/s62kv/.

## Abstract

Can people categorize complex visual scenes unconsciously? The possibility of unconscious perception remains controversial. Here, we addressed this question using psychophysical methods applied to unmasked visual stimuli presented for extremely short durations (in the μsec range) by means of a custom-built modern tachistoscope. Our experiment was composed of two phases. In the first phase, natural or urban scenes were either absent or present (for varying durations) on the tachistoscope screen, and participants were simply asked to evaluate their subjective perception using a 3-points scale (absence of stimulus, stimulus detection or stimulus identification). Participants' responses were tracked by means of two staircases. The first psychometric function aimed at defining participants' proportion of subjective detection responses (i.e., not having seen anything vs. having seen something without being able to identify it), while the second staircase tracked the proportion of subjective identification rates (i.e., being unaware of the stimulus' category vs. being aware of it). In the second phase, the same participants performed an objective categorization task in which they had to decide, on each trial, whether the image was a natural vs. an urban scene. A third staircase was used in this phase so as to build a psychometric curve reflecting the objective categorization performance of each participant. In this second phase, participants also rated their subjective perception of each stimulus on every trial, exactly as in the first phase of the experiment. Our main result is that objective categorization performance, here assumed to reflect the contribution of both conscious and unconscious trials, cannot be explained based exclusively on conscious trials. This clearly suggests that the categorization of complex visual scenes is possible even when participants report being unable to consciously perceive the contents of the stimulus.

## Introduction

Can we categorize complex visual scenes even if we report being unaware of the content of the scene or that a stimulus was present? Today, this fundamental question remains controversial.

**Funding:** This work was supported by an European Research Council Advanced Grant RADICAL to Axel Cleeremans (Grant ERC-2013-ADG #340718). The funders had no role in study design, data collection and analysis, decision to publish, or preparation of the manuscript.

**Competing interests:** The authors have declared that no competing interests exist.

It is still unclear whether perceptual processes can take place without awareness [1–3]. However, different lines of research, extending over a century, have suggested that perception without awareness is actually possible.

Beyond the early work of Sidis [4] and Stroh and collaborators [5], Marcel [6,7] was one of the first authors to explore the possibility of unconscious perception in a set of experiments involving visual masking in word recognition paradigms. More specifically, he showed that participants could accurately match in meaning and in shape a word masked by a subsequent visible word, even when the first word was reported as not having been consciously seen. In addition, in a color identification experiment, when the masked name of a color was congruent with the color of a subsequent patch, participants' identification reaction times decreased. Conversely, when the masked color name was different from the color of the patch, their reaction times increased. This phenomenon, known as a *priming effect* ([8]; but see [9,10] for reviews), strongly suggests that visual information can be computed even for unconscious information, and that it can influence participants' behaviour. Later, Cheesman and Merikle [11,12] introduced important terminology that is still used today and that distinguishes between objective and subjective thresholds. In their experiments, the authors showed that when reducing the temporal interval between a prime and a mask, participants reported guessing about the presence of the prime. However, at this same threshold, the prime's influence on the objective task (and adapted Stroop task) was significantly above chance level. Thus it appears that there is a first threshold in terms of stimulus energy (i.e., signal strength), above which we are able to perform better than chance on a given task while remaining unaware of the stimulus (i.e., objective threshold), and a second threshold, higher in stimulus energy, above which we are aware of seeing something (i.e., subjective threshold).

As indicated above, the most convincing evidence for unconscious perception comes from priming experiments. However, theories of consciousness such as the Global Neuronal Workspace Theory [13,14], High-Order Theories [15,16] or the Self-Organizing Metarepresentational Account [17] all support the idea that unconscious perception should be observable through direct measures, that is, without relying on indirect methods such as priming. Indeed, these theories all assume that consciousness specifically depends on mechanisms operating "above" the mechanisms that drive first-order perception. Thus, in Global Neuronal Workspace Theory, unconscious information in a perceptual module in the brain becomes available to consciousness when this information enters the workspace and can henceforth be shared with other modules in the brain. Hence, unconscious perception is clearly possible at the level of individual processors, as long as the computed information does not enter the workspace, so triggering its "ignition". Higher-Order Theories (HOT) take it as a starting point that unconscious information becomes conscious only when it is the target of a high-order thought, assumed here to involve wholly distinct processes. Adopting a similar view, the Self-Organizing Metarepresentational Account (SOMA) proposes that second-order meta-representations are necessary to enable a first order perceptual representation to become available for consciousness. In short, both HOT and RPT theories suggest that the extent to which a first-order representation is conscious depends on its being indexed by a second-order (unconscious) representation. Crucially however, the first-order processes are assumed to take place regardless of whether they are being monitored by this second-order processes.

All these theories under consideration here would predict that direct unconscious perception is possible. Thus far however, the possibility of such direct unconscious perception has been received with skepticism, with some authors even claiming that unconscious perception does not exist ([1–3]; but see [18,19], amongst many others, for recent counter-evidence).

We think that the main reason for this skepticism stems from the substantial methodological challenges associated with demonstrating unconscious perception [20] and with the

differences in the specific ways in which the studies have been carried out and their results interpreted. Thus, for instance, the definition of subjective and objective thresholds, *per se*, varies highly from study to study. Indeed, authors can either choose the objective or the subjective threshold as the threshold of consciousness. While the objective threshold, by definition, can be objectively defined, there are many experimenter degrees of freedom when choosing a subjective threshold (for a deeper description of the subjective/objective threshold problem, see [1]). Thus for instance, a conservative experimenter could define the subjective threshold as the point of energy below which participants fail to categorize a single stimulus as conscious on a subjective scale. A liberal experimenter, on the other hand, may choose as the threshold the point of energy where some stimuli are categorized as conscious (half of them, for example). While such differences in experimental design hinder the comparison between different studies, they also fail to capture the fact that thresholds are most likely not static fixed points in stimulus energy— they vary not only across individual participants, but also over the entire distribution of stimulus energies. This state of affairs mandates an approach that take the dynamics of perceptual decision making into account, that is, an approach that examines the entire distribution of subjective responses as a function of stimulus energy. This, we believe is the only approach that makes it possible to take both properties of the stimulus as well as individual participants' own internal states into account to define the shape of transition between unconscious and conscious processing.

Koch and Preuschoff [21] already suggested a somewhat similar theoretical approach (for an application, see [22]). They suggested to approach the question of establishing the extent to which task performance is driven by unconscious processing through the comparison of psychometric functions defined over different stimulus energy levels. A first psychometric function represents objective performance on a given task, while a second represents perceptual awareness as expressed through a subjective scale such as wagering or visibility judgments. Comparing the functions can then reveal a "lack of consciousness". For instance, observing that the subjective psychometric is shifted on the x-axis compared to the objective psychometric is suggestive that subjective reports lag objective reports, which the authors claim may be interpreted as revealing unconscious processing [22]. While the proposed method is a very interesting departure from more traditional methods to assess unconscious processing, we also think that it fails to be sufficiently precise. Merely comparing a *qualitative* measure of consciousness and a *quantitative* measure of performance lacks common metrics. Further, interpreting the observed patterns of association and dissociation between the two measures may also lack sufficient sensitivity. For instance, it may be the case that the observed objective performance associated with a given level of stimulus energy stems exclusively from a few trials ranked highly on the awareness scale (i.e., conscious trials).

Here, we suggest a novel method that crucially involves comparing *quantitative* estimates of both objective performance and subjective report. While task performance can easily be objectively assessed through measures such as reaction time or accuracy, obtaining quantitative measures of subjective reports is challenging. One way of addressing this challenge is to use psychophysical staircases to track, as objectively as possible, the subjective thresholds at which participants transition from 1) absence of experience to 2) subjective stimulus *detection* and to 3) subjective stimulus *identification* (Fig 1). The resulting subjective psychometrics can then be interpreted as representing what we call the complete subjective threshold of participants, namely, the likelihood, at each stimulus energy level, that participants use different points of the scale. Note that this is very different from the typical subjective psychometric functions reported in the literature (e.g., [22]). Whereas those functions simply plot how, on average, participants use the different points of a subjective scale, our psychometric functions represent, for each stimulus energy level and each participant, the proportion of trials that

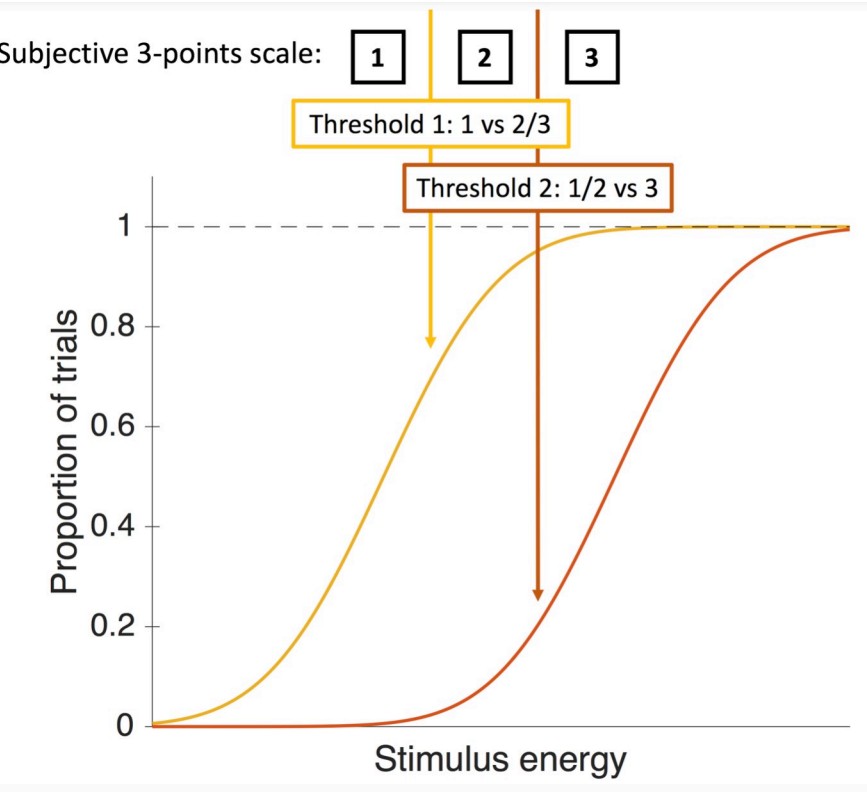

**Fig 1. Evolution of subjective thresholds with stimuli energies.** A 3-points subjective scale is taken as example. The participant's use of the scale can be precisely tracked by psychometrics representing the likelihood of crossing different perceptual thresholds described through the scale.

were subjectively detected or identified. Importantly, this depends on the scale's sensitivity to clear differences in qualitative perception. We can then precisely study how objective and subjective reports are linked.

Another important idea raised by Koch and Preuschoff [21] is that the extent to which the transition between unconscious and conscious processing is gradual can be assessed through the slope of the subjective psychometric function. Thus, a shallow slope indicates that participants use the entire scale and is suggestive that consciousness is graded, whereas a sharp, stepwise transition can be interpreted as indicating that consciousness is all-or-none. However, we believe this interpretation is not warranted, essentially because it is impossible, with current methods, to distinguish between all-or-none awareness of graded contents and graded awareness *tout court*. Indeed, if we examine the extant literature on this point, while some studies have shown that responses on a subjective scale are gradual [23], others have not ([24]; but see [25–27], for a mixed view), leaving the question open. The question is even more complex. For example, Sandberg and Overgaard's 4-point Perceptual Awareness Scale (PAS [23]; see also [28] for the seminal paper) aims at measuring participants' subjective visibility as a mirror of their (graded) experience of the stimulus. The labels on the scale are "no experience", "brief glimpse", "almost clear experience" and "clear experience". However, when a participant selects "brief glimpse", it is still not clear from the scale whether he experienced degraded consciousness of the stimulus, or whether he was fully conscious that his perception of the stimulus was blurry. While the first interpretation suggests that conscious access to a given content is intrinsically graded, the second interpretation suggests that the graded nature of the reports

stems from all-or-none access to graded contents. In other words, it seems essential to dissociate the contents of consciousness from conscious access in itself [29] and to specify, for each, whether they operate in an all-or-none or in a graded manner.

Here, in the absence of consensus, we have adopted the perspective that conscious access is all-or-none because we believe that being conscious of more is different from being more conscious [29]. Hence, whatever the process that brings information to consciousness, this information reaches the threshold of consciousness or fails to do so in a dichotomous way. Information in itself could undoubtedly be degraded, due for instance to low stimulus energy or to noise in perceptual information processing. In short, we assume in this article that we either have some subjective contents of a stimulus or we do not, independently of the fact that this subjective content is more or less rich than in another situation, condition, or person.

The assumption we made about consciousness in our approach is reminiscent of similar ideas raised by the partial awareness hypothesis [30,31]. This theory proposes that the different levels of representation associated to a given stimulus are separately available to consciousness in an all-or-none fashion. For example, if we present a word to participants, they might have conscious access to level of energy, features and letters of the word, but not to its meaning. In such a case, participants would report having perceived some letters while remaining unable to identify the meaning of the word. We introduce the same kind of conscious differentiation in our scale. Subjective stimulus *detection* is comparable to awareness of the first level of representation of the stimulus (its energy level), while subjective *identification* is comparable to conscious access to the highest level of representation of the stimulus (the nature of the visual scene). Following the partial awareness hypothesis, finer-grained scales could be used to find additional thresholds of consciousness for different intermediate levels of representation.

In this study, we therefore explored the distribution of responses on the subjective scale for each stimulus duration, through psychometric fitting, and compared it to objective responses. This methodology allowed us to precisely define the consciousness threshold not only in terms of stimulus energy, but by also taking into account each participant's sensitivity dynamically. We rendered complex visual stimuli unconscious by significantly reducing the duration they were presented on the screen of a modern tachistoscope, enabling μ-seconds stimulus presentation with high accuracy. The tachistoscope enables greater precision and better control over stimulus duration, thus alleviating the pitfalls of masking methods [1,20,32]. Previously, Sperdin and colleagues [33] used a similar device and were able to find brain activation for very short stimulus durations, even under subliminal conditions. However, we still do not know if this information allowed participants to perceive a specific stimulus unconsciously. Our experiment was divided into 2 phases. For the first phase, the stimulus was either absent or present for durations varying between 0 and 15000μs, as determined through two simultaneous staircase procedures aimed at fitting both subjective detection and subjective identification. In this first phase, participants were simply asked, on each trial, to judge the visibility of the stimulus on a 3-point scale. For the second phase, participants were instructed to categorize the same stimuli as *natural* or *urban* by pressing the left arrow or right arrow keys respectively. On each trial, after categorizing the stimulus, participants were asked to perform the same subjective task as in Phase 1 again. Stimuli were again presented for a range of durations, this time driven by a staircase procedure applied to participants' objective responses.

Our main hypothesis is that participants should be able to exhibit better-than-chance objective performance even if stimulus energy does not reach the subjective threshold of consciousness. But disentangling conscious from unconscious performance has been shown to be a difficult task, particularly if one wants to measure a level of unconscious performance. We solved this problem by tilting the way of addressing it: to measure unconscious performance,

we want to show that performance on the conscious trials cannot explain the objective performance of the experiment alone.

## Materials and methods

### Participants

Forty-five participants were recruited for this experiment (36 females; 40 right-handed; mean age and standard deviation: 20.2 ± 3.6 years). All were French speakers, with normal or corrected to normal vision and students in Psychology at the Université libre de Bruxelles (ULB, Belgium). They participated in exchange for course credit. Informed consents were obtained from each participant and the experiment was conducted in a properly ethical manner in agreement with the Declaration of Helsinki (2008). The present study was specifically approved by the Ethics Committee of the Erasme Hospital (P2017/563).

### Material

Stimuli were natural and urban scenes selected from a database created by Oliva and Torralba [34]. A hundred and sixty "open country" and 160 "highway" pictures were selected from the sample. For the "open country" pictures, we selected only those without any human presence on them. The pictures were 256 x 256 pixels large, black and white images controlled for their global luminance and RMS contrast thanks to the SHINE toolbox running on Matlab [35].

### Tachistoscope

We used a custom-made LCD tachistoscope to display the stimuli. Details about the device can be found in the Appendix (for an older version of the tachistoscope, see [33,36]. In short, the tachistoscope is composed of two LCD screens reflecting on a single semi-transparent mirror, which allowed to show stimuli at the microsecond level (below 16ms of presentation, precision of 2μs; see Appendix). Given the testing distance of 36 cm (distance between participants' eyes and the screen), stimuli subtended 10 x 10 degrees of visual angle.

### Procedure

The experiment lasted for 2 hours and was divided into 2 phases of one hour each. Instructions were given to the participants at the beginning of each phase.

For the first phase (648 trials), the stimulus, flanked by 2 blank screens (Fig 2), was either absent (17% of trials = 108 trials) or present for durations varying between 0 and 15000μs (= 15ms; 83% of the trials = 540 trials) through a staircase procedure (see Staircases). Participants were asked, on each trial, to judge the visibility of the stimulus on a 3-point scale (see objective and subjective tasks for a description of the scale). There was no objective task associated to this phase.

For the second phase (492 trials), the same stimuli were used and presented between 2 blank screens at the same durations as for Phase 1. The same proportion of present/absent trials was used as in Phase 1. That is, the stimulus was absent on 82 trials and present, at different durations, on 410 trials. When the stimulus was present, 78% of trials (320 trials) were the actual "test" trials. The remaining 22% (90 trials) were used as "catch" trials to evaluate whether there was a shift in participants' subjective responses over the course of the experiment, by comparing with Phase 1, as an influence of the objective task [37]. The presentation time for catch trials was calculated from Phase 1's participant responses, for both the subjective detection and the subjective identification thresholds (see Staircases). For this phase, participants were also instructed to categorize the stimulus as *natural* or *urban* by pressing the left

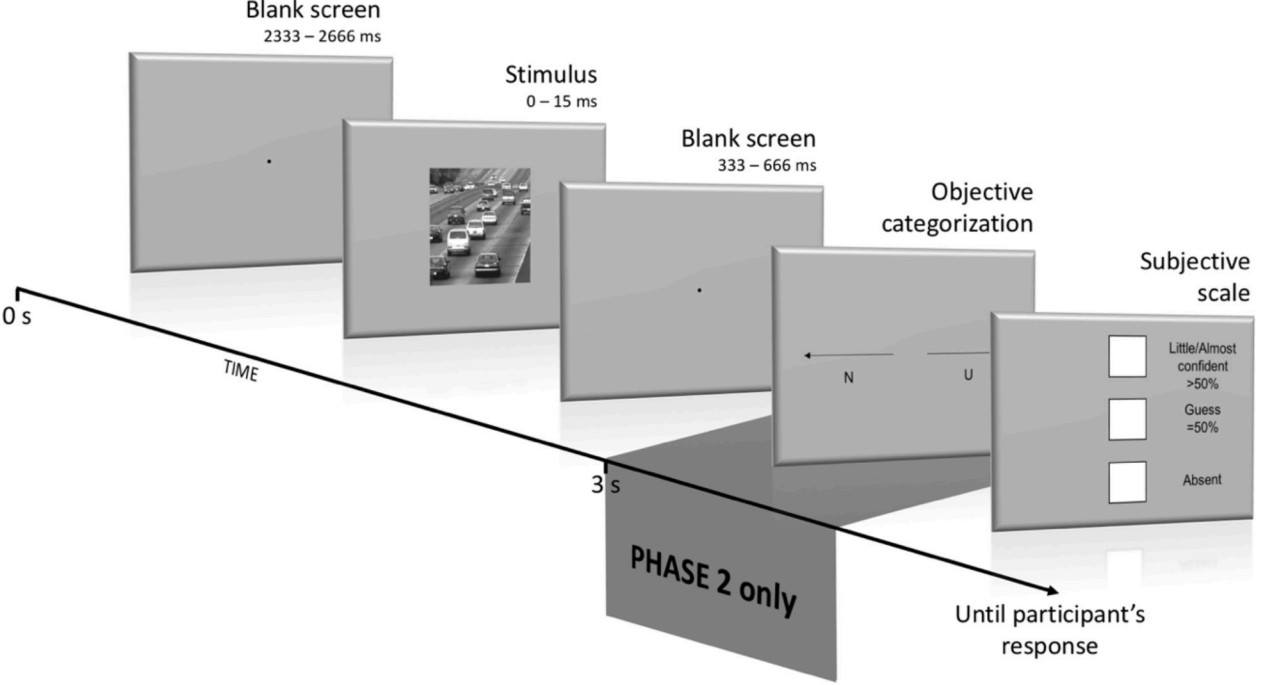

**Fig 2. Schematic view of the experimental design.** Phase 1 only required a *subjective* judgment from participants whereas Phase 2 included an additional *objective* categorization task, the goal being here to explore whether an objective categorization response can have an influence on a subjective judgment.

arrow or right arrow keys respectively. After categorizing the stimulus, participants had to perform the subjective task and were asked, as such, to judge the visibility of the images on the same 3-point scale as used in Phase 1 (Fig 2).

## Objective and subjective tasks

For the *objective* task, participants were asked to categorize each image as belonging to either a "natural" or an "urban" scene by pressing the corresponding "left" key or "right" key of the keyboard. To prompt participants to respond, two arrows appeared on the screen. One arrow pointed leftwards to a "N" letter, and the other pointed rightwards to a "U" letter. Participants had no time pressure to respond.

For the *subjective* task, we used a three-points PAS scale similar to the one used in Sandberg et al. [22], which combined the "almost clear experience" and the "clear experience" option from the original scale [23]. This 3-point scale was presented vertically on the screen to avoid any possible motor contamination originating from an earlier phase (i.e., objective on subjective for Phase 2). Participants used the up and down arrows to navigate the scale as well as the space bar to validate their choice, again without any time pressure. Critically, participants were instructed to respond "absent" if they thought that no stimulus was presented on the screen (they were told in advance that this could be the case for some trials). If they detected a stimulus but could not categorize it as either a natural or an urban scene and they only had the impression to guess on this, they were asked to select the "guess" option. Finally, if participants felt they could identify the stimulus (whatever their confidence was), they were asked to choose the "little/almost confident" option. By using such a scale, we aimed at targeting two distinct subjective capabilities, each one referring to distinct conscious experiences a subject can have. The first one is subjective *detection* (stimulus vs. noise), which refers to participants' subjective

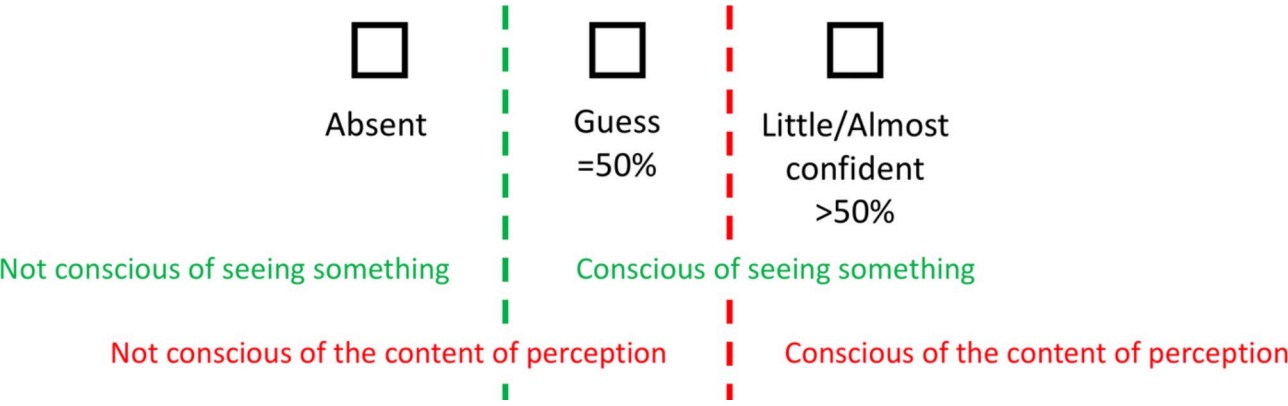

**Fig 3. Subjective 3-point PAS scale and what it refers to in terms of consciousness.**

judgments of what lies between "absent" and "guess". In between these two judgments lies the threshold between "not conscious of seeing something" and "conscious of seeing something" (Fig 3). The second is *subjective identification* and refers to what lies between "guess" and "little/almost confident" to participants. In between lies the threshold between "not conscious of the content of the stimulus" and "conscious of the content of the stimulus" (Fig 3).

## Staircases

We used staircases to reach very specific stimulus duration for each individual. Hence, we were able to compare, for each participant, the level of performance on the objective task and conscious perception of stimuli. Three distinct staircases were used over the entire experiment.

For Phase 1, there were 2 intermixed converging staircases, each one applied on 270 trials (a stimulus was present on all these trials). The goal of the first ("staircase 1") and second ("staircase 2") staircase was to find the presentation time at which participants could subjectively report detecting and identifying 50% of the stimuli, respectively. To do so, the "detection" staircase used reports on the first point on the subjective scale as indicating "false" (i.e., stimulus absent) and the two others as indicating "true"; while the "identification" staircase used the first two responses as indicating "false" and the third response as indicating "true".

For Phase 2, we used a third staircase ("staircase 3") composed of 320 test trials that aimed to find the stimulus duration for which participants are 75% correct at the objective categorization task.

We used the PSI method and the Palamedes Matlab toolbox [38] to compute the staircases. This entropy reduction method makes it possible to achieve precise convergence of the staircase for each participant. Another advantage of the PSI staircase is that it uses the data to find both the threshold and the slope parameters of the best-fitting psychometric curve (in this case, a cumulative normal function).

The psychometric function we obtained for each participant on Phase 2 thus represents performance in objective categorization as a function of the time of presentation (Fig 4A, 4B or 4C; blue curve). The psychometric functions we obtained on Phase 1 represent the proportion of subjectively detected/identify trials for each stimulus duration (Fig 4.A, 4B or 4C, red curve).

## Analyses

We started from the assumption that conscious access is dichotomous (see Introduction). As defined earlier, we also distinguished two types of subjectivity: an all-or-none access to

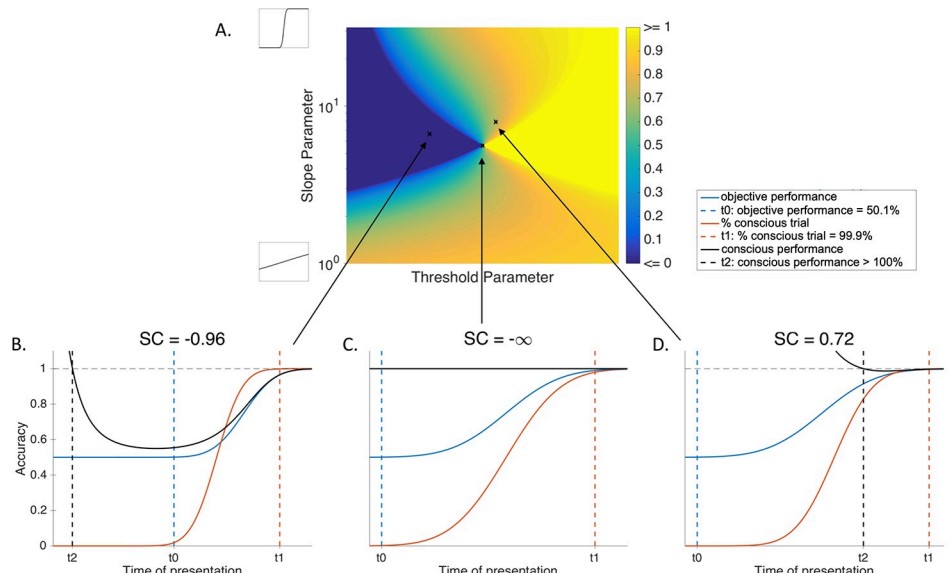

**Fig 4. SC for different subjective psychometric parameters.** Simulated data; stimulus duration and threshold parameters do not correspond to real times. A. Topology of SC for different parameters of participants' subjective psychometric curves. Each pixel combines two parameters: threshold and slope. SC could either be negative (blue) or positive (yellow) depending on participants' subjective psychometric parameters, given some fixed couple of objective psychometric parameters. The center point corresponds to objective psychometric parameters. Black curves on the left correspond to the theoretical shape the psychometric curve would show given the minimum or the maximum slope parameter. Threshold parameters affect only the position of the psychometric on the x-axis. B. Simulation of a negative SC. In this case, all objective performance (blue curve) between t0 (objective performance at chance) and t1 (all trials are conscious) is explained by conscious performance (black curve) on conscious trials (red curve). C. Theoretical case of an ideal observer, where objective and subjective thresholds are the same. Subjective parameters are the same as objective parameters. In this case, SC cannot be calculated with formula (4), but it theoretically converges to -∞. Participants have 100% performance on conscious trials and 50% performance on unconscious trials. D. Simulation of a positive SC. In this case, part of objective performance between t0 and t1 cannot be solely explained by conscious performance on conscious trials.

subjective detection and an all-or-none access to subjective identification (See objective and subjective tasks). Based on these definitions, we assumed that all trials in the experiment were associated to subjective experience, or not, in an all-or-none fashion.

Hence, performance at the objective task at a given duration is calculated as the sum of performance on conscious and unconscious trials. This idea is operationalized through the following equation:

$$OP = \%C * CP + \%U * UP \tag{1}$$

OP: objective performance, %C: percentage of conscious trials, CP: conscious performance, %U: percentage of unconscious trials, UP: unconscious performance.

Since each trial is categorized as either being conscious or unconscious, the proportion of unconscious trials can be directly explained according to the proportion of conscious trials. As such, the Eq (1) becomes:

$$OP = \%C * CP + (1-\%C) * UP \tag{2}$$

In our study, OP was extracted from individual psychometric curves fitted based on their objective categorization responses. %C was instead calculated from their psychometric curves fitted on their subjective identification responses (or subjective detection responses, depending on which staircase we are looking for). Crucially, we also assumed that unconscious

performance should be set at chance level, i.e. UP = 0.5 (See Introduction). Given these assumptions, the Eq (2) now becomes:

$$CP = (OP - (1-\%C) * 0.5)/\%C \qquad (3)$$

Hence, in our analysis, we verified whether conscious performance could, on its own, explain objective performance. If not, the hypothesis that unconscious performance is at chance level becomes false. Thus, the effect we want to measure is the proportion of objective performance (Fig 4, blue curve) that cannot be explained exclusively based on participants' performance on conscious trials (Fig 4, black curve). We will refer to this as the *surplus coefficient (SC)*.

To measure SC at an individual level, three values were extracted from the fitted psychometric curves. The first corresponds to the stimulus duration that was necessary for accuracy to be at 50.1%, t0. This value indeed corresponds to "chance level" and objective performance below this value is mostly explained by 50% performance on both conscious and unconscious trials. Then we measured the stimulus duration for which participant's subjective psychometric curves reaches 99.9%, t1. This point corresponds to the threshold at which participants are conscious of almost all (99.9%) trials. Consequently, objective performance is explained exclusively by performance on conscious trials. Finally, we measured the point in time where conscious performance would (absurdly) need to be higher than 100% to explain objective performance, t2, based on Eq (3). SC is therefore computed as such:

$$SC = (t2 - t0)/(t1 - t0) \qquad (4)$$

Theoretically, SC cannot be negative. It can only be null (if objective performance is exclusively explained by conscious trials) or positive (if conscious trials cannot explain all of the objective performance). However, because of the limitation of the mathematical tools we used here, the objective psychometric is never perfectly equal to 50%. We therefore fixed a boundary below which we considered that SC is mostly due to noise in the mathematical calculation of effect, compared to the real effect in the data. That would occur when all of objective performance between t0 and t1 can be explained by performance on the conscious trials, except when time of presentation is exceptionally small (objective performance smaller than 50.1%, see Fig 4B). This threshold of 50.1% prevented us from having only a positive SC due to the formula. Hence, as one can notice in Fig 4A and 4B, SC could be negative if t2 < t0.

## Results

### Data pre-processing

We excluded 7 participants from the sample because they did not perform the task adequately on one of two criteria (5 and 1 participant excluded because of criterion 1 or criterion 2, respectively), or both (1 participant excluded). Criterion 1 targeted those participants who incorrectly identified the absence of the stimulus (i.e., reported it to be present despite it was not) on at least 60% of the trials in which no stimulus was shown, on both Phases 1 and 2 (See Procedure). Criteria 2 targeted those participants in Phase 2 who were below 60% correct in the objective classification task when responding with the highest point on the subjective scale (i.e. "little/almost confident") (See Objective and subjective tasks). The threshold of 60% was fixed a priori as a conservative criterion to be sure that the measure is different from chance level (50%).

## Objective performance *vs.* subjective response in phase 2

In Phase 2, both objective and subjective responses were recorded on each trial, which allowed us to assess performance on the objective task as a function of subjective categorization on the 3-point subjective scale (Fig 5). When participants responded "absent", mean objective performance was 0.50 (not different from chance level, one-sample t-test: t = 0.09; p > 0.05; Bu[0, 0.13] = 0.154, if we use mean "guess" response as a plausible maximum effect [39]. When participants responded "guess", their mean objective performance was 0.63 (t = 10.56; p < 0.001; Bu[0, 0.43] >> 100, if we use mean "little/almost Confident" response as a plausible maximum effect). Finally, when participants responded "little/almost Confident", their mean objective performance was 0.93 (t = 59.08; p < 0.001; Bu[0, 0.5] >> 100, if we use maximum performance as a plausible maximum effect). That is, objective categorization takes place even in the absence of subjective content identification. However, it seems that when people do not know that something is presented, they cannot perform better than chance.

We further took advantage of psychometric curves fitted on staircase data to reveal the dynamics of consciousness thresholds (see Method - Staircases). In particular, we associated participants' subjective detection and identification rates to their objective categorization rates with the goal of precisely determining whether participants' subjectivity can explain objective categorization, at different levels of energy of the stimulus. To do so, we first compared participants' psychometric curves.

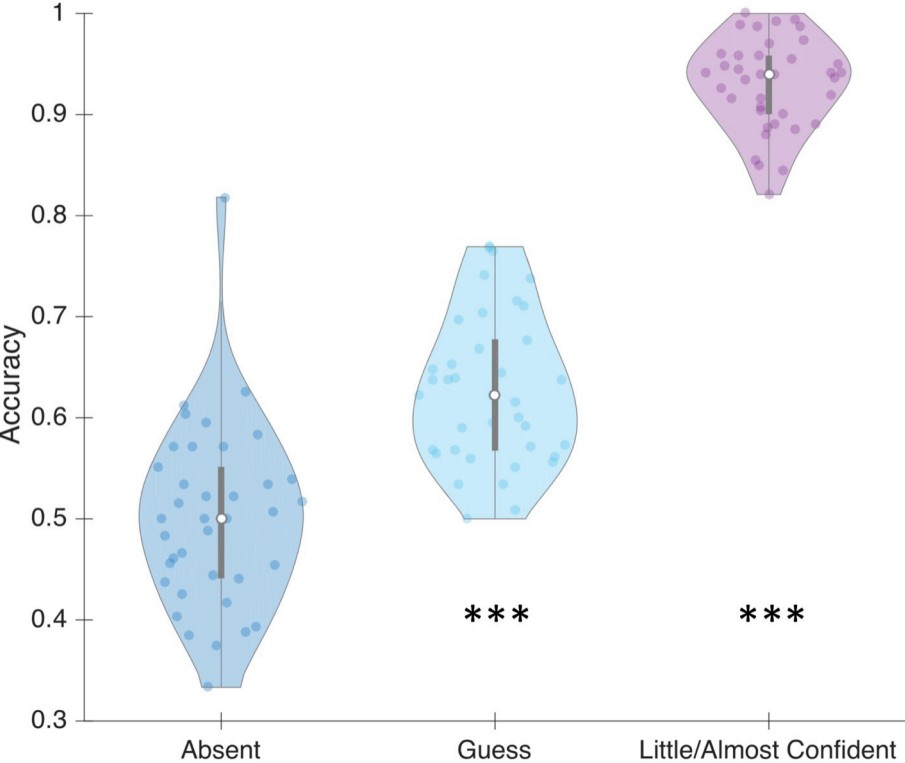

**Fig 5. Each individual objective categorization score plotted as a function of subjective rating (absent, guess or little/almost confident) in Phase 2.** Dots represent individual values. Bars represent different quartiles. Envelopes indicate the density distribution of the participants.

## Comparison of psychometric curves

Three psychometric curves were fitted out of participants' staircases (Fig 6; see Method - Staircases). The first one characterized their proportion of subjective detection ("staircase 1"), the second one their subjective identification ("staircase 2") and the third one their objective categorization scores ("staircase 3") (Fig 7). The parameters of the psychometric functions can be seen in Fig 8 for individual participants, and in Table 1 the group medians.

## Surplus Coefficient (SC) calculation

As explained in Method (Analyses), SC corresponds to the proportion of objective performance that cannot be explained by conscious trials under the hypothesis that performance is exclusively driven by conscious contents. We computed it for both subjective detection and identification (see Fig 9). At the group level, subjective detection was not significantly different from 0 (median = 0.110; Wilcoxon test: z = -0.102; p = 0.540) whereas it was for subjective identification (median = is 0.599; Wilcoxon test: z = 5.134; p < 0.001). That is, objective performance can be only fully explained by performance on the trials where participants report having detected the stimuli. The same is not true for when they report having identified the stimuli.

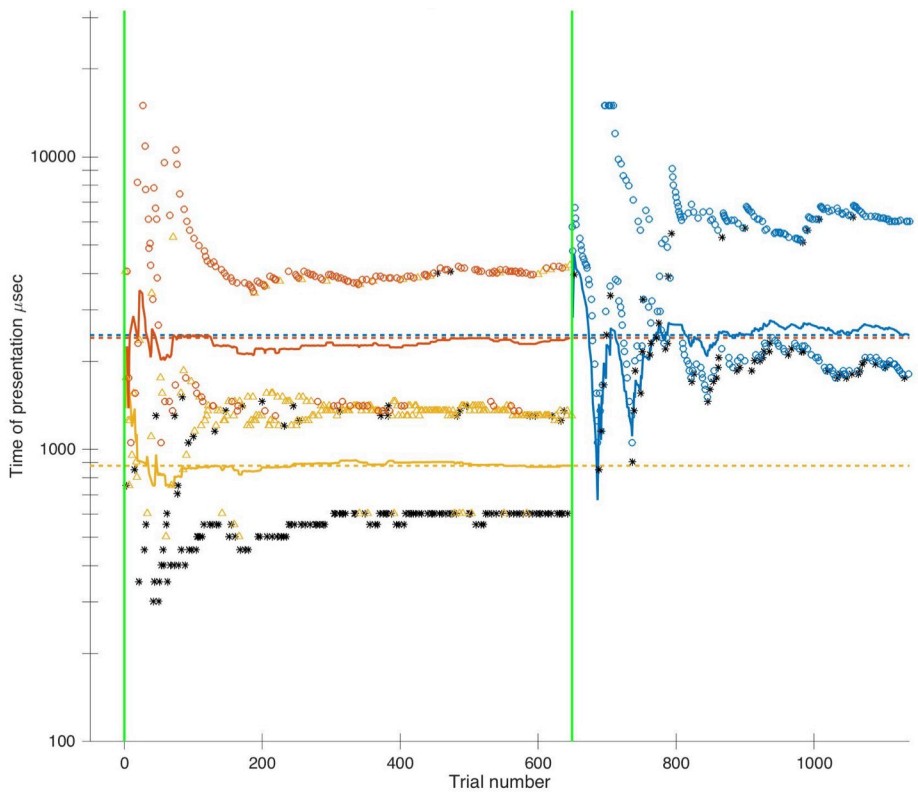

**Fig 6. Time course of the 3 staircases over the course of the experiment for one participant.** Trial number is displayed on the horizontal axis and stimulus duration on the vertical axis. Phase 1 is delineated between the 2 vertical green lines. Phase 2 begins right after. For phase 1, each dot represents one trial and therefore a stimulus. It is a black star when it has been categorized as "absent", a yellow triangle when categorized as "guess" and a red circle when categorized as "little/almost" confident. In addition, the yellow and red curves indicate how the threshold parameter converge for the subjective detection psychometric and the subjective identification psychometric, respectively. For Phase 2, black stars and blue circles are indicative of trials not categorized correctly, and correctly categorized in the objective categorization task, respectively. The blue curve indicates convergence of the threshold parameter for objective categorization task.

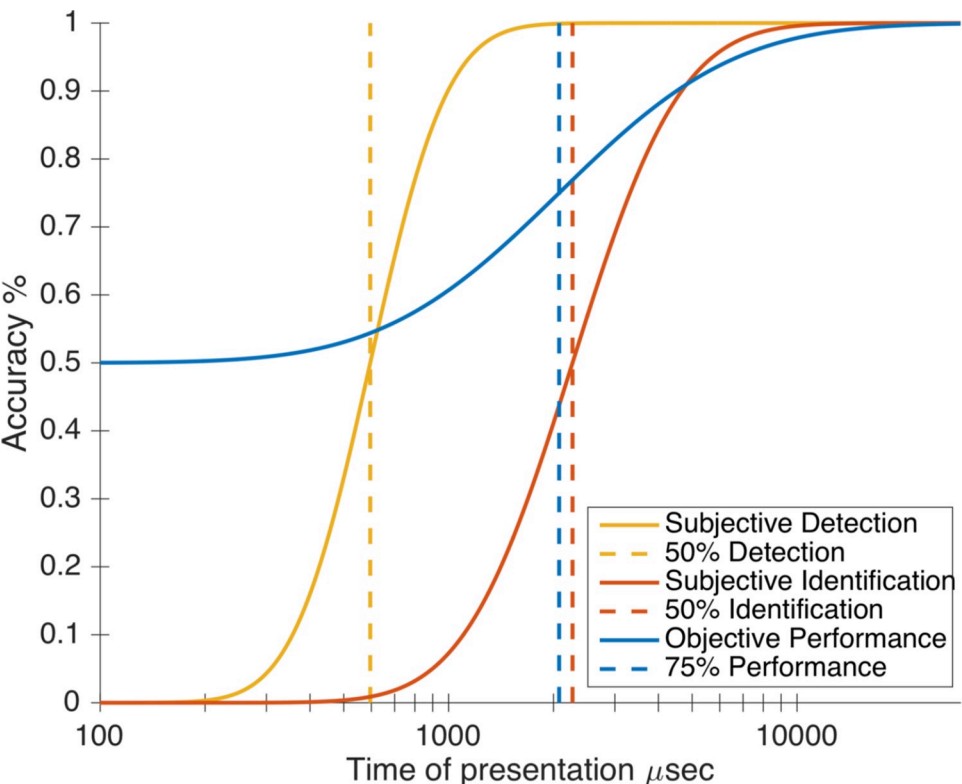

**Fig 7. Psychometric curves based on the global parameters (i.e. median of participants' parameters, see Table 1).** Stimulus duration is represented on the x-axis. The blue curve represents objective performance across participants for the objective categorization task (Phase 2). The yellow curve represents the proportion of trials reported as having been subjectively detected. The red curve represents the proportion of trials reported as subjectively identify.

To control for whether this pattern of results is due to a potential response bias (i.e., participants using the visibility scale in a liberal or conservative fashion), we performed additional signal detection theory (SDT) analyses [40,41], in Phase 2. Indeed, both objective and subjective responses were recorded on the same trial. We calculated for each participant, second order Hit and False Alarm rates [41]. The Hit rate was computed by dividing the proportion of correct trials at objective categorization when participants reported that they could subjectively identify the stimulus by global accuracy. The False Alarm rate was computed by dividing the proportion of incorrect trials on the objective categorization task when participants reported they could subjectively identify the stimulus by the global error rate.

These two measures were then introduced in an analysis to estimate the area under the ROC curve ([42]; see also [43] who used the same methodology). For each participant, the ROC curve is composed of three points: (0,0), (Hit rate, FA rate) and (1,1). A first estimation of the area in the triangle (0,0), (Hit rates, FA rates) and the point formed by the orthogonal curve to the diagonal, passing by (Hit rates, FA rates), is given by the formula below [42]:

$$K'_A = \tfrac{1}{4}(h-f)\left(h+f+\tfrac{f}{(1-f)}\right);$$ where h is Hit rate and f False Alarm rate

Then, the second triangle between (Hit rates, FA rates), the point formed by the orthogonal curve to the diagonal and (1,1) is calculated with:

$$K'_B = \frac{1}{4}(h-f)\left(2 - (h+f) + \frac{(1-h)}{h}\right);$$

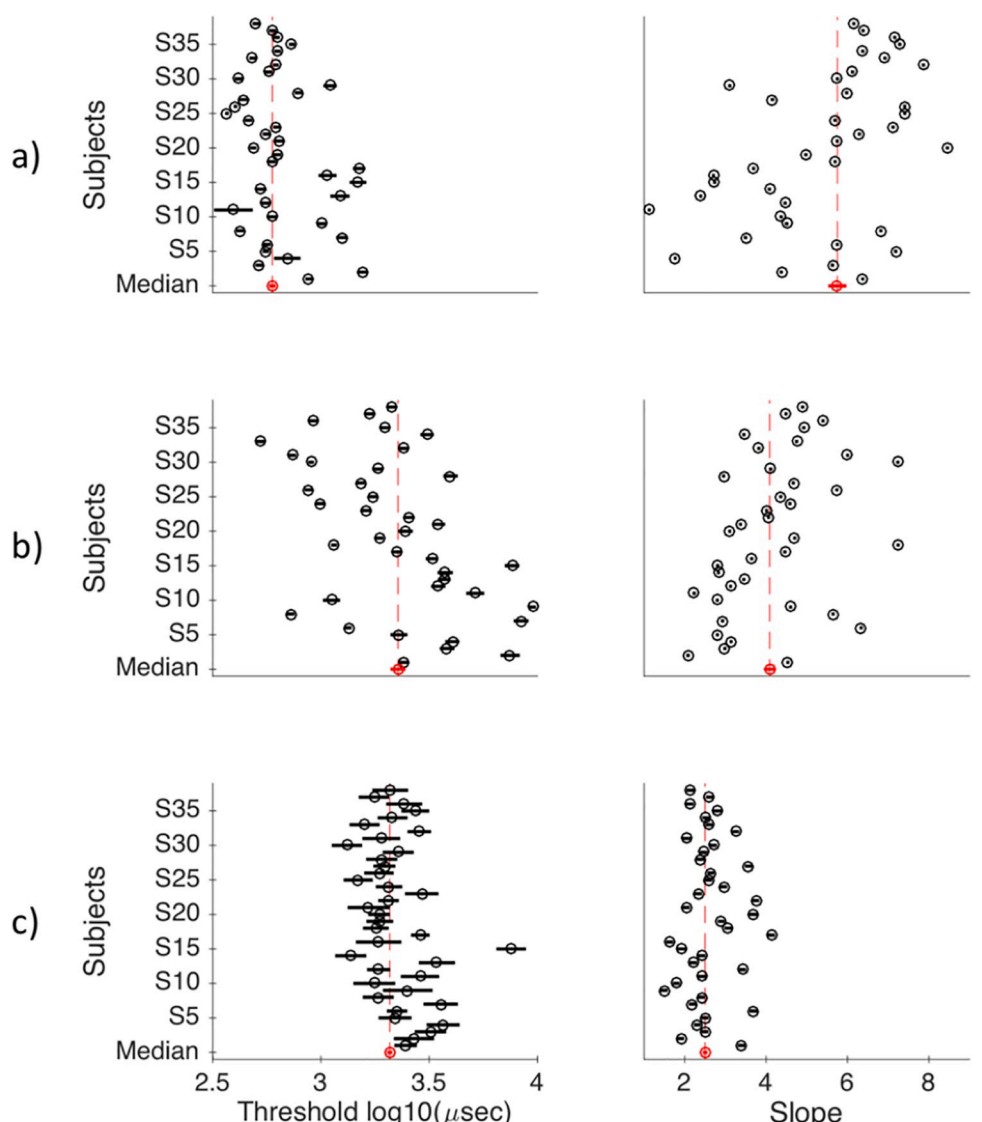

**Fig 8. Psychometrics' parameters for all participants.** (a) subjective detection psychometric; (b) subjective identification psychometric and (c) objective categorization psychometric. Black circles represent the exact value of the threshold/slope parameter and black lines represent the standard error. The red circle at the bottom represents the median of all parameters and red lines represent median absolute deviation (MAD) divided by the square root of the total number of participants.

**Table 1. Psychometrics' global parameters.**

| Psychometric curve | Parameter | Participant parameters | |
|---|---|---|---|
| | | **Median** | **MAD** |
| Subjective detection | Threshold | 2.775 | 0.089 |
| | Slope | 5.750 | 1.374 |
| Subjective identification | Threshold | 3.356 | 0.216 |
| | Slope | 4.090 | 0.896 |
| Objective categorization | Threshold | 3.317 | 0.072 |
| | Slope | 2.499 | 0.372 |

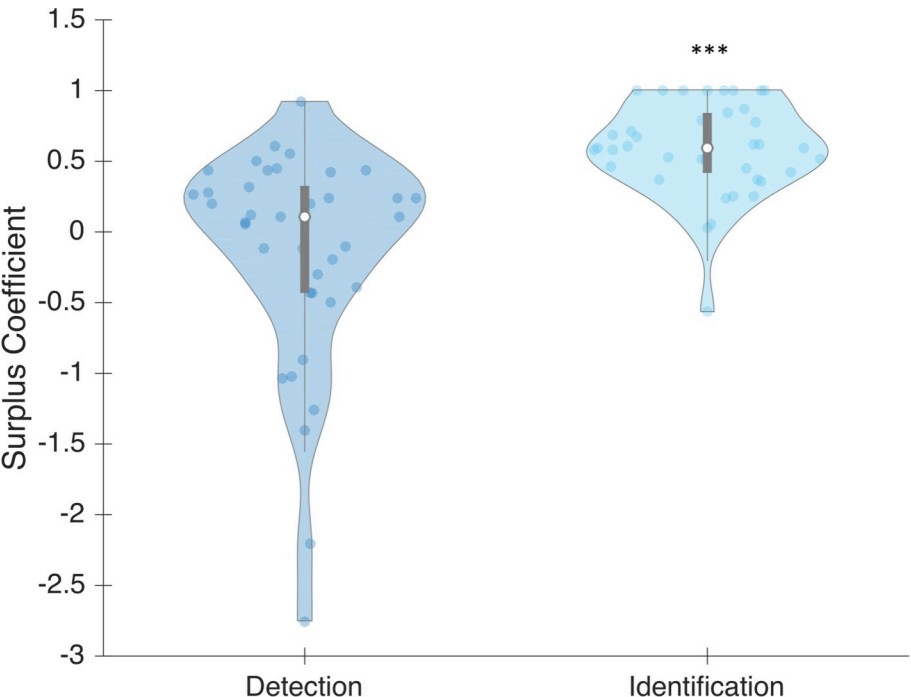

**Fig 9. SC plotted as a function of the subjective categorization used (subjective detection or subjective identification) in Phase 1.** Dots represent individual values. Bars represent different quartiles. Envelopes indicate the density distribution of the participants.

Based on these two equations, the sensitivity measure $A_{ROC}$ and Type II bias $B_{ROC}$ are defined as follows:

$$A_{ROC} = K'_A + K'_B + 0.5;$$

$$B_{ROC} = ln\left(\frac{K'_A}{K'_B}\right);$$

to determine whether participants are rather conservative or liberal when responding on the subjective scale (Fig 10.A).

As a second step, we fitted a subjective identification psychometric on visibility responses in Phase 2. Then, we performed a linear regression on this data to determine whether the bias could predict SC. This was indeed the case (F(2,36) = 22.8, p < 0.001), with an $R^2$ of 0.387. Participants' SC is equal to 0.536–0.196 * Bias. Hence, even if SC is influenced by bias, SC remains even with no bias (Fig 10B), because the intercept found in the regression is positive.

As we wanted to know which parameters, namely the threshold or the slope, of the psychometric is the most impacted by participants' bias, we performed as a third step, 2 simple linear regressions. We first predicted the threshold based on bias. There was a significant regression (F(2,36) = 98.1, p < 0.001), with an $R^2$ of 0.732. Participants' predicted threshold is equal to 3.284–0.279 * Bias. The threshold is thus strongly associated to participant bias (Fig 10C), suggesting that the psychometric threshold is linked to the criterion in signal detection theory. In SDT, bias affects only the criterion, and not sensitivity. To verify this second point, we predicted the slope parameter based on bias. The slope parameter is indeed strongly linked with participant sensitivity. As expected, no significant regression was found (F(2,36) = 3.32, p = 0.077), with an $R^2$ of 0.084.

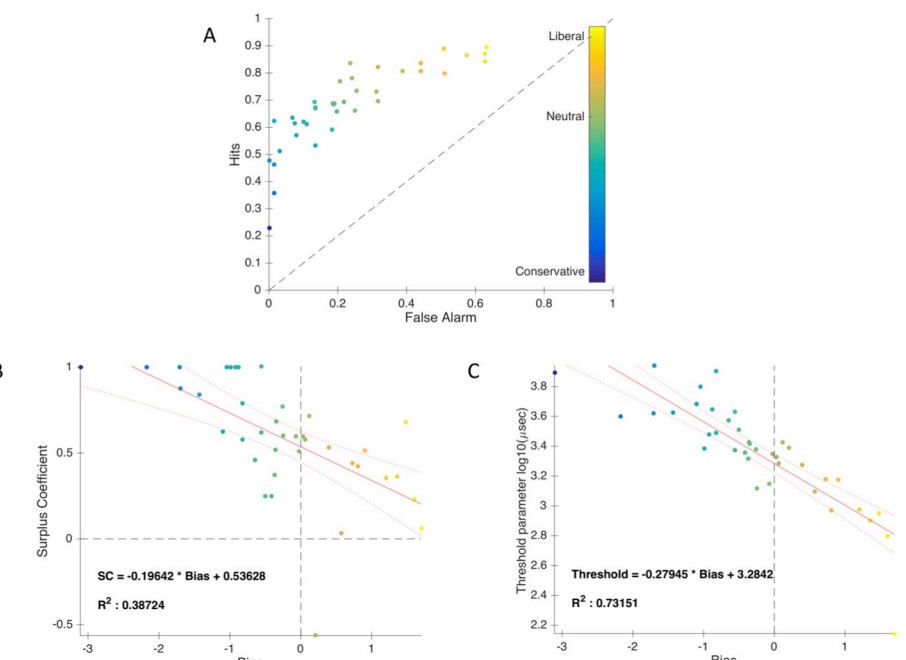

**Fig 10. Metacognitive bias.** A. Proportion of hits (y-axis) and false alarms (x-axis) for each participant. The color of the dots indicates whether participants exhibit a bias in their subjective responses compared to their objective performance. B. Regression of SC as a function of bias for all the participants. Color of the dots indicate participant's bias. C. Regression of the threshold parameter in function of Bias for all participants. The color of the dots indicates the participant bias.

With the regression of *threshold* compared to *bias* (Fig 10C), it is possible to determine an ideal non-biased value of *threshold* parameter, which is the intercept. This enabled us to investigate further the individual influence of sensitivity on SC. To achieve this, we used the threshold of 1924 µsec ($10^{3.29}$ in Fig 10C), the point at which non-biased participants could theoretically identify subjectively half of the trials. We used this threshold for all participants, without changing participant sensitivity (slope parameter). Even if the global SC was reduced after correction (0.664 after bias correction versus 0.825 before; $z = 1.791$, $p = 0.073$), it remained significant ($z = 3.34$, $p \ll 10-3$) (see Fig 11).

Based on the calculation of the bias for Phase 2, we inferred the bias for Phase 1, in which only subjective ratings were recorded. As depicted in Fig 12, this comparison led to the observation that the threshold parameter was not different ($z = -0.472$, $p = 0.637$), contrary to the slope ($z = 4.793$, $p \ll 10-3$). This finding suggests that participants kept the same subjective response criterion through both phases. In contrast, the change in the slope parameter indicates that response sensitivity varies throughout the different phases, an argument in favor of the idea that producing an objective response influences (reduces, in this case) subjective sensitivity.

Given that bias only affects the criterion, it seems reasonable to assess that bias in the first phase is the same as bias in the second phase. We applied the same technique as before to measure SC based only on sensitivity. We replaced all individual threshold parameters by the non-biased threshold that we identified previously, but kept participants' individual slope parameter. We found that the global SC is the same after correction (0.623 after bias correction versus 0.599 before; $z = 0.877$, $p = 0.38$) (see Fig 13). After bias correction, SC remains significant ($z = 4.322$, $p \ll 10-3$).

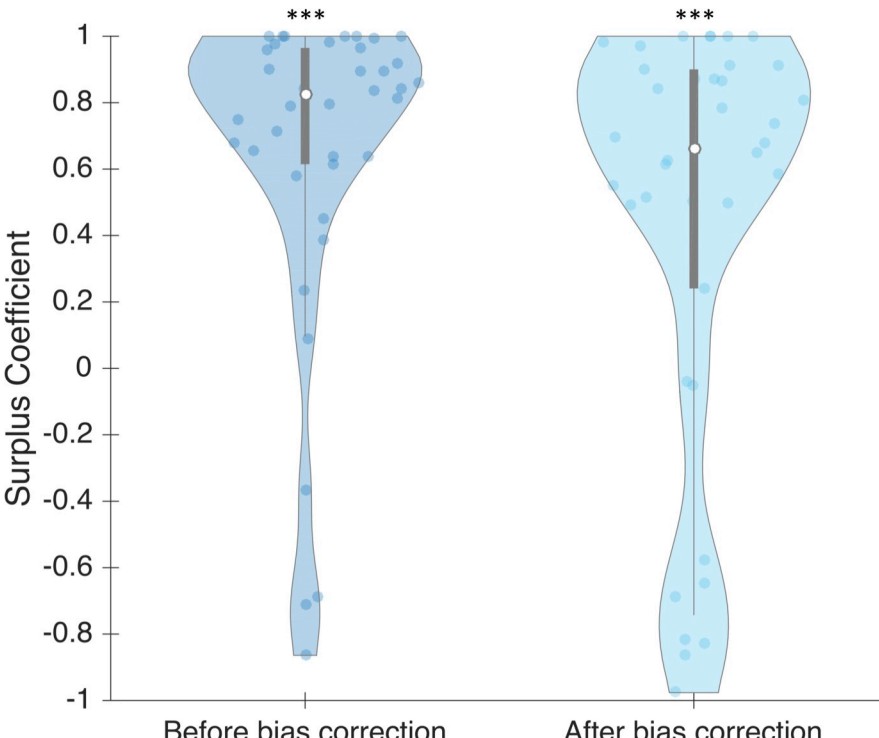

**Fig 11. SC for each participant before and after bias correction, in phase 2.** Dots represent individual values. Bars represent different quartiles. Envelopes indicate the density distribution of the participants.

## Discussion

The goal of this experiment was to establish whether stimuli that people report being unable to identify can nevertheless be correctly categorized. To do so, we looked at the entire dynamics of the subliminal threshold for each participant individually, using a new design of adaptive staircases for both objective and subjective reports. Participants were presented with visual scenes, and asked 1) to rate their visibility on a 3-point scale (subjective task) and 2) to categorize the scene as being urban vs. natural (objective task). We aimed to show that objective performance in this task cannot be entirely accounted for based on conscious processing. To do so, we assumed that performance can indeed be entirely accounted for by conscious processing [1], and showed that it cannot (reductio ad absurdum).

Consistent with our main prediction, conscious trials alone were not sufficient to explain objective performance on the categorization task. The proportion of trials for which participants reported being able to identify the content of the visual scene was not sufficient to explain global objective categorization over all trials. In other words, when participants reported being unaware of the content of the stimulus ("absent" or "guess" responses), they could still perform better than chance on the objective categorization task. By *reductio ad absurdum*, we conclude that direct unconscious perception [20] is necessary to explain objective performance in our task, and more precisely that objective identification appears to be possible without subjective identification. However, our results fail to show the necessity of unconscious categorization when participants report being unaware that something was presented on the screen ("absent" responses): objective identification appears to be impossible when the participant failed to detect the stimulus—a pattern that is the landmark of the "blindsight effect" [44–49]. Indeed, for unconscious performance to be observable in our task,

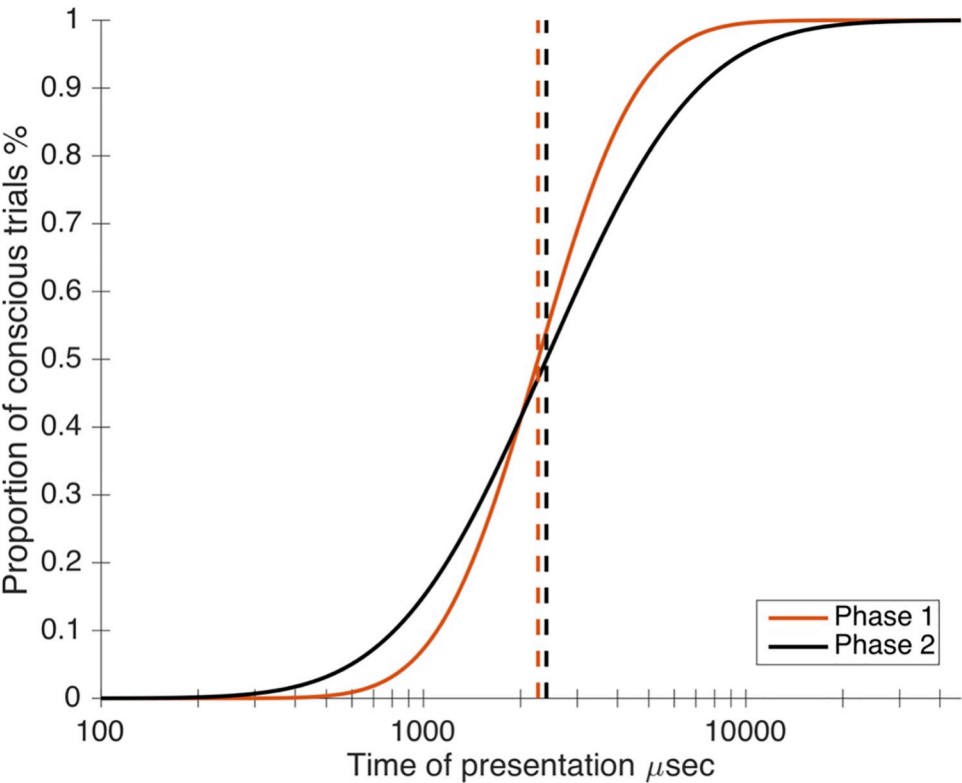

**Fig 12. Global subjective identification recorded in phase 1 and in phase 2.** The threshold parameter is similar between the two phases (Phase 1: 3.356; Phase 2: 3.381) while the slope differ (Phase 1: 4.090; Phase 2: 2.716).

participants needed to know (as indicated by either "guess" or "little/almost confident" reports) that something was presented on the screen.

Another important result is that subjective reports were less sensitive when participants were required, in Phase 2, to carry out both objective and subjective tasks, suggesting that performing the objective task biases subjective responses. One possible explanation for this loss of sensitivity is the increased cognitive demand associated with dual task performance: in the second phase, participants presumably needed to exert increased cognitive effort to maintain information in working memory until producing their subjective report. Alternatively, we cannot rule out a possible learning effect, since trials that only involved the subjective task always occurred in the first phase of the experiment. However, we note that such a learning effect would predict the opposite effect: if participants indeed learned to respond more accurately using the subjective scale, we would expect to find greater, rather than smaller, sensitivity in the second phase of the experiment.

The existence of direct unconscious perception is widely debated in the current consciousness literature, with evidence for [19] and against [1,3]. The debate has been partly fed by the lack of consensus about the appropriate methodology to test unconscious perception. Consequently, there has been disagreement among researchers on the definition of what constitutes the subliminal threshold for consciousness. Response bias is particularly problematic when using subjective scales, as conservative participants, who tend to underreport their subjective experience, may lead one to conclude for unconscious processing when there is only response bias. It is thus necessary to control for this possible bias, and if found, to apply a procedure to correct for it. Our new method made it possible to show that the *Surplus Coefficient* (SC; see

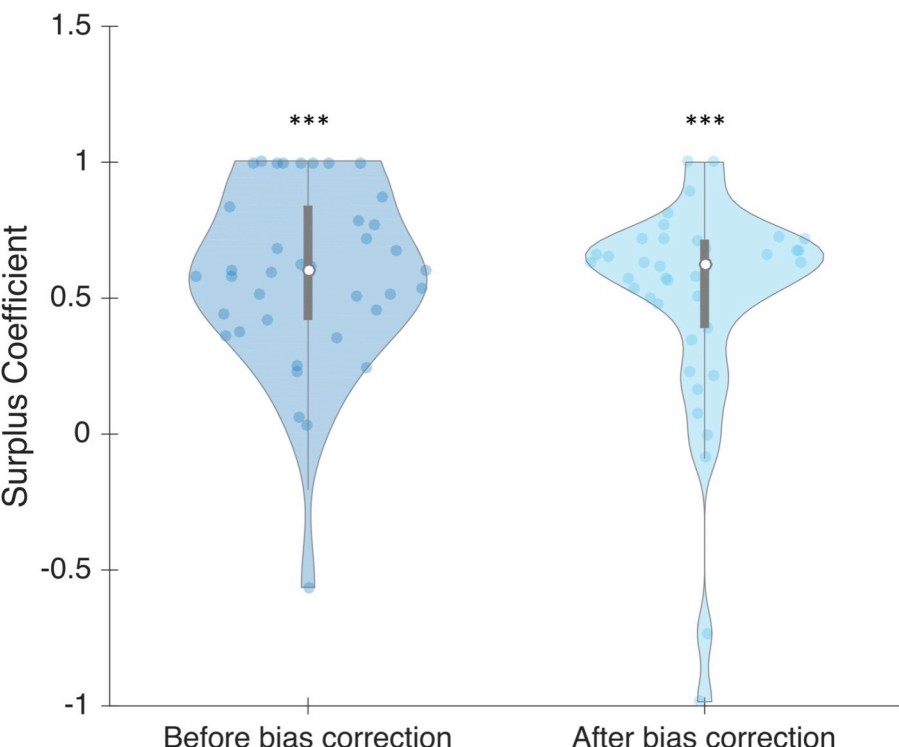

**Fig 13. SC for each participant, before and after bias correction, in phase 1.** Dots represent individual values. Bars represent different quartiles. Envelopes indicate the density distribution of the participants.

Method) was present even after controlling for each participant's own bias and only examining their sensitivity. Moreover, our methodology to detect unconscious performance was meant to be very conservative: we wanted to determine whether performance on conscious trials is sufficient to explain objective performance, a hypothesis that we have disproved. Indeed, finding that in some situations performance on conscious trials cannot explain all the global objective performance, even if performance on conscious trials were 100%, does not seem realistic. Despite the conservative position and methods we adopted, our results show that performance on subjectively identified trials was not sufficient to explain the objective performance on the objective categorization task, favoring the interpretation that unconscious performance is strongly present. However, a possible limitation of our method is that participants' subjective responses could depend on the specific images we used (i.e., participants' subjective detection and their identification thresholds may be different for each image). In this article, we assumed that the possible variability in the subjective thresholds stemming from image-specific factors is lower compared to the variability stemming from participants own global thresholds because images were matched on both luminance and contrast. However, we cannot exclude a possible influence of picture on subjective thresholds.

Another advantage of our method is the use of novel technology to present sub-millisecond stimuli without having to rely on masking. Masking is a technique that allows suppressing content from conscious processing, and consists of showing a stimulus (a "mask") immediately after a target stimulus, therefore rendering it invisible. Using masking inevitably involves noise in perception, a possible confound which was avoided here by using the tachistoscope without masking.

Overall, the findings of this study are twofold. First, we showed that decisions about the content of a stimulus are possible even when participants fail to be able to identify those very

contents, hence demonstrating direct unconscious perception. Second, our results suggest that subjective judgments, when collected concurrently with objective responses, are less sensitive than when collected alone. While we cannot be certain that this loss of sensitivity stems from interference or from cognitive fatigue, this result has important implications for the design of studies aimed at studying unconscious perception, since many relevant studies have now espoused this methodology.

## Supporting information

**S1 File.**
(DOCX)

**S1 Fig. Illustration of our tachistoscope design.** a) A semi permeable mirror is positioned between two monitors arranged in a 90˚ configuration. One of the monitor's light is reflected (screen 1), while the second's passes through the mirror (screen 2). b) The screens and mirror are supported by a rigid structure in aluminium. c) The complete device is covered with black plexiglass. An optional front plate with a small aperture can be added to ensure optimal head position and minimal room light interference.
(TIF)

**S2 Fig. The tachistoscope has a high temporal precision.** Here are examples of stimuli ranging from *20* μ*s* to *1000* μ*s* (1*ms*). The luminosity of each screen is measured separately and we observe that the tachistoscope is precise to 1μ*s*. We also observe a small transitional phase of 2μ*s* caused by the switch of monitors.
(TIF)

**S3 Fig. Detailed measurements for a stimulus of 500 micro-seconds.** Switch between the two screens: Resulting luminosity is a combination of the two screens and do not produce a visible variation of luminosity. To display a stimulus, the main screen (screen 1) is powered off while screen 2 is powered on. After the stimuli duration, screen 2 is powered off and screen 1 powered on. The "computed" curve (in black) is a result from addition of screen 1 and screen 2, while the "measured" curve (in green) is obtained experimentally directly on the tachistoscope.
(TIF)

**S4 Fig. Tuning of the tachistoscope.** Tuning of the screen luminosity. Because of the mirror asymmetry and the LCD difference, the perceived luminosity of the screen can vary (black). Using a rheostat in series with the screen blacklight, tuning is performed to set both screens at a similar value (blue). The spikes correspond to the switch between the 2 monitors. Calibration was performed on a 500μ*s* stimulus.
(TIF)

## Acknowledgments

We are very grateful to Jérôme Sackur, Bert Timmermans, Zoltan Dienes, Maxime Maheu and Laurène Vuillaume for their insightful comments on previous versions of the present article, and to Wafae El Hammouchi for her support in data collection.

## Author Contributions

**Conceptualization:** Arnaud Beauny, Adélaïde de Heering, Santiago Muñoz Moldes, Jean-Rémy Martin.

**Data curation:** Arnaud Beauny.

**Formal analysis:** Arnaud Beauny.

**Funding acquisition:** Axel Cleeremans.

**Investigation:** Arnaud Beauny.

**Methodology:** Arnaud Beauny.

**Project administration:** Arnaud Beauny.

**Resources:** Arnaud Beauny.

**Software:** Albert de Beir.

**Supervision:** Axel Cleeremans.

**Validation:** Axel Cleeremans.

**Visualization:** Arnaud Beauny.

**Writing – original draft:** Arnaud Beauny, Albert de Beir.

**Writing – review & editing:** Arnaud Beauny, Adélaïde de Heering, Santiago Muñoz Moldes, Jean-Rémy Martin, Axel Cleeremans.

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
