## [Decision Letter · Decision Letter 0]

3 Dec 2019

PONE-D-19-20674

Unconscious categorization of sub-millisecond complex images

PLOS ONE

Dear Mr. Beauny,

Thank you for submitting your manuscript to PLOS ONE. After careful consideration, we feel that it has merit but does not fully meet PLOS ONE’s publication criteria as it currently stands. Therefore, we invite you to submit a revised version of the manuscript that addresses the points raised during the review process.

All three reviewers found your manuscript interesting and novel. However, they have also found a number of methodological issues that need to be addressed before the paper is ready for publication. Enclosed are the views of the experts to whom I turned for evaluation of your manuscript (one of them is Julian Matthews).

One particularly important issue is data availability – if you could share the data as a supplementary file or deposit it on a one of the research data sharing platforms, that would be great.

We would appreciate receiving your revised manuscript by Jan 17 2020 11:59PM. To enhance the reproducibility of your results, we recommend that if applicable you deposit your laboratory protocols in protocols.io, where a protocol can be assigned its own identifier (DOI) such that it can be cited independently in the future. For instructions see: http://journals.plos.org/plosone/s/submission-guidelines#loc-laboratory-protocols

We look forward to receiving your revised manuscript.

Kind regards,

Magdalena Ewa Król, Ph.D.

Academic Editor

PLOS ONE

Journal Requirements:

3. We note that Figure 2 in your submission contain copyrighted images. All PLOS content is published under the Creative Commons Attribution License (CC BY 4.0), which means that the manuscript, images, and Supporting Information files will be freely available online, and any third party is permitted to access, download, copy, distribute, and use these materials in any way, even commercially, with proper attribution. For more information, see our copyright guidelines: http://journals.plos.org/plosone/s/licenses-and-copyright.

1.    You may seek permission from the original copyright holder of Figure 2 to publish the content specifically under the CC BY 4.0 license.

Additional Editor Comments (if provided):

Reviewers' comments:

Reviewer's Responses to Questions

**Comments to the Author**

1. Is the manuscript technically sound, and do the data support the conclusions?

Reviewer #1: Partly

Reviewer #2: Yes

Reviewer #3: Partly

2. Has the statistical analysis been performed appropriately and rigorously? 

Reviewer #1: Yes

Reviewer #2: Yes

Reviewer #3: I Don't Know

3. Have the authors made all data underlying the findings in their manuscript fully available?

Reviewer #1: No

Reviewer #2: Yes

Reviewer #3: No

4. Is the manuscript presented in an intelligible fashion and written in standard English?

Reviewer #1: Yes

Reviewer #2: Yes

Reviewer #3: Yes

5. Review Comments to the Author

Reviewer #1: The authors report the findings of a psychophysical experiment aimed at assessing unconscious perception of complex visual scenes. Critically, the authors employ a tachistoscope capable of precise, sub-millisecond stimulus presentation, and outline a novel method for inferring unconscious perception by 'reductio ad absurdum' that combines perceptual awareness ratings and psychometric threshold comparison.

Their method takes as a starting point that objective performance on a task should be explainable by performance on only those trials that a participant reports being conscious (on the conservative assumption that unconscious performance is at chance-level). The authors apply this logic to a novel analysis technique for inferring conscious performance on the basis of subjective and objective performance thresholds (“surplus coefficient” or SC). Using SC, the authors reveal that the conservative position that unconscious performance is at chance-level cannot hold. Critically, “the proportion of trials for which participants reported being able to identify the content of the visual scene was not sufficient to explain global objective categorization over all trials.”

In general, the authors do a fantastic job of outlining the existing literature in their introduction, the assumptions their study adopts, and the novelty their method brings. The authors employ a tachistoscope to target two types of conscious perception: detection and identification. This procedure allows them to psychophysically distinguish two thresholds of visual awareness with respect to urban and natural scenes. Firstly, the threshold between "not seeing" and "seeing", and secondly, the threshold between "not conscious of content" and "conscious of content". I believe this procedure is a worthwhile contribution to this literature. However, I have reservations about the other major element of their paper—the novel method for inferring unconscious perception. This is why I consider the data to only partly support their conclusions.

Major issue:

My concern is the derivation of “surplus coefficient” (SC). The goal of this analysis is to determine the theoretical level of performance on “conscious” trials that would be required to explain “objective performance”, assuming chance-level performance on “unconscious” trials. The upshot of this process is that when SC is greater than 0, it is impossible to explain “objective performance” on the basis of conscious trials alone because “conscious performance” needs to be greater than 100% accurate. This logic seems to hold in the context of this experiment when we consider conscious “detection”. This is because the target of a detection report is the mere perception of a stimulus—we can assume conscious performance will be roughly equivalent for all stimuli on conscious trials. This is not the case for conscious “identification”. In this context, the target of the report is one of 160 urban or 160 natural scenes. It is very likely that this judgment is easier or harder for certain stimuli, even under the assumption that conscious access is a dichotomous process.

Given this, it seems mistaken to assume “conscious performance” will be fixed for all conscious trials (and vice versa for “unconscious performance”) when deriving SC (“we verified whether conscious performance could, on its own, explain objective performance”). Despite this concern, I think it’s unlikely stimulus variability entirely discounts the authors’ findings in the “identification” condition. However, it will affect the replicability of this study and extensions were a different set of stimuli used. This issue should be addressed when discussing the results and when using the phrase “direct unconscious perception” in their Discussion - it seems SC , in its present form, is not quite appropriate in the case of conscious identification.

Stimulus variability is a common issue in the psycholinguistic literature that is accounted for by treating stimuli as a random factor in the error term of a mixed model. This approach may be appropriate here, for instance modelling objective performance as a function of psychometric thresholds, slopes, and conscious and unconscious performance as well as including stimuli and subjects as random factors. Another advantage of this approach is that it allows the inclusion of all trials in the model, and would be robust to changes in the proportion of conscious and unconscious trials + changes in threshold as the experiment progresses.

Minor issues:

As per PLOS ONE guidelines, “contacting author” is not sufficient data availability. I recommend uploading the data to Figshare in .csv format along with a ‘readme’ .txt document that describes each variable.

I was a bit uncertain about the response process and timing in Phase 2, especially for stimulus “present” trials. My understanding is that participants made a “natural” vs. “urban” judgment and a subjective report on every trial (i.e., “absent”, “catch”, and “test” trials). The critical difference was that on “catch” trials the presentation time was determined from responses in Phase 1. But was this the time corresponding to “detection” or “identification” reports?

Line 201 “same stimuli stimulus” —> “same stimuli”

Line 307 “detecting and identify” —> “detecting and identifying”

Line 308 “on the subjective as” —> “on the subjective scale as”

Line 387 “Data are simulating” —> “Simulated data”

Line 399 “same than objective” —> “same as objective”

Line 412 “below than 60% correct” —> “below 60% correct”

Line 421 “was of 0.50” —> “was 0.5”

Line 515 “and of false” —> “and false”

Line 538 “enabled to investigate” —> “enabled us to investigate”

Reviewer #2: This paper describes a psychophysical experiment on unconscious categorization of complex visual images (natural vs. urban scenes) presented with exposure durations in the submillisecond range using a custom built LCD-tachistoscope. Subjective perception (detection/visibility) was evaluated in a first phase (no objective task), followed by a second phase (objective task) where participants had to decide what the stimulus was on each trial (forced choice categorization). In this second phase, subjects had also to perform after each trial the same subjective task as in the first phase (absence of stimulus, stimulus detection or stimulus identification). The authors explored the distribution of responses on the subjective scale for each stimulus exposure duration, through psychometric fitting, and compared it to objective responses.

Results indicate that participants are able to correctly decide about the content of a stimulus (that is, to correctly categorize it) even when they fail to identify the contents per se, thus suggesting direct unconscious perception (for this specific setup using complex visual scenes).

This study provides interesting novel results that have implications on the understanding of conscious and subliminal perception when using complex visual stimuli. The paper is well written, and the Methods and Results are thoroughly described.

I think the discussion could be improved by considering other results from other papers that have used a similar approach to render the stimuli subliminal. There is at least one other paper (to my knowledge) that has used a similar device (https://onlinelibrary.wiley.com/doi/10.1002/hbm.22716) with submillisecond exposure duration. In this experiment the authors used rather simple visual stimuli (checkerboards) and found that although the checkerboards were undetectable on a conscious level for the 250 microsecond duration (see their experiment 3), yet they still evoked a brain response (even if a small one only). The authors should relate their present findings to this, at least with respect to the method used.

Reviewer #3: This study presents an experiment in which subjects need to subjectively evaluate the visibility (phase 1) of natural or urban scenes. In a next phase, the need to objectively categorize these scenes and also give a subjective visibility evaluation. The surplus coefficient is introduced as new measure to explain the proportion of objective performance that cannot be explained by the subject’s performance on conscious trials only. The results show that subjects can classify scenes even when they fail to subjectively fail to see the contents of the stimulus.

Overall, I found this an interesting study to read with some intriguing results. However, I am not convinced that the use of the surplus coefficient is bringing a lot of novelty or a better measure to the field of consciousness research. I listed my comments below.

- I find I it hard to fully appreciate the surplus coefficient (SC) as the dependent measure. While I get the intuition of the SC, some aspects still remain obscure to me after several readings of the analysis paragraphs. I understand that what is called global performance in Figure 5 is the objective performance, and that the % conscious trials in the figure corresponds to the subjective performance (it would help if there is more consistency in the naming). If I understand correctly, the counterintuitive suggestion of an objective performance that needs to be higher than 100% is needed to fit equation (3) and because unconscious performance (UP) is assumed to be equal to 50% in this equation. When looking at the simulations, however, I get the impression that the authors could use a simpler measure than this. If it is assumed that consciousness is an all-or-none phenomenon, the threshold for subjective/objective consciousness can be calculated as the inflexion point of the logistic regression curve (see Del Cul et al., 2007, PLoS Biol). According to my understanding of the surplus coefficient it should be highly similar to the difference in these thresholds: If the objective threshold is lower than the subjective threshold, this suggests that there is some additional – unconscious - processing taking place without reaching the subjective threshold (or a positive SC; Fig. 5D). In the case of an ideal observer (Fig.5C), the objective threshold is equal to the subjective threshold (the difference between the two is 0; or the SC is undetermined), suggesting that there is no additional unconscious processing. In the unlikely case of a negative SC (Fig. 5B), the subjective threshold is lower than the objective threshold which is also highly unlikely as it would mean that subjects can’t objectively classify information that they subjectively label as conscious. The two measurements (difference between thresholds and the SC) seem to behave the same. Because the use of the SC as a new metric is one of the most important claims of this study, it would be nice to understand why the authors constructed this new measure rather than to directly compare the thresholds and why is it beneficial to use the SC. The results with the SC appear to be very much in line with more traditional analysis approaches (e.g., aren’t the results presented in Fig. 6 giving you the exact same conclusions about unconscious processing?).

- In the discussion, the authors write that their results show that unconscious performance is only observable when participants knew there was something on the screen (lines 592-597). This statement appears to be rather paradoxical - a conscious sensation is necessary to have unconscious processing? – and needs to be rewritten. Also, later in the discussion it is written that the current results strongly favor unconscious performance what seems to be contradicting what was written before. The authors need to spell out more in detail what they exactly refer to: Objective identification appears to be possible without subjective identification, but is impossible without subjective detection, or something like this.

- What is missing from the experiment is an objective detection task. It would have been informative to see how this thresholds relates to the subjective detection threshold and the objective identification threshold.

- Abstract, line 31: “without being to identify it”. Missing word.

- Page 9, line 201: “…stimulus stimulus…”. Remove stimulus.

- Page 19, line 412: “… who were below than 60%...”. Remove than.

- Page 19, line 420-421: “…mean objective performance was of 0.50…”. Remove of.

- Figure caption of figure 9 says that the yellow curve represents the proportion of correct responses on trials reported as having been subjectively detected. Is this not just the proportion of trials indicated as subjectively seen? Same for the red curve.

- Page 26, line 593:”…our results fail to show the necessity of unconscious categorization…”. Strange sentence.

- Page 26, line 600: It is concluded that subjective reports were less sensitive in phase 2 what would suggest that performing the objective task biases subjective responses. However, only a change in the slope was observed and no change in the threshold. Does this then reflect a change in bias?

- Page 27, line 623: “…a hypothesis that we infirm.” Should be confirm.

- Page 27, line 626: “…that performance on subjectively identify trials..”. Should be rewritten.

6. PLOS authors have the option to publish the peer review history of their article (what does this mean?). If published, this will include your full peer review and any attached files.

Reviewer #1: Yes: Julian Matthews

Reviewer #2: No

Reviewer #3: No

---

## [Author Response · Author response to Decision Letter 0]

23 May 2020

PONE-D-19-20674

Unconscious categorization of sub-millisecond complex images

Response to referees

In the following, we address each referee’s point. Referee comments appear in roman typeface; our responses appear in italics.

Reviewer #1: The authors report the findings of a psychophysical experiment aimed at assessing unconscious perception of complex visual scenes. Critically, the authors employ a tachistoscope capable of precise, sub-millisecond stimulus presentation, and outline a novel method for inferring unconscious perception by 'reductio ad absurdum' that combines perceptual awareness ratings and psychometric threshold comparison. Their method takes as a starting point that objective performance on a task should be explainable by performance on only those trials that a participant reports being conscious (on the conservative assumption that unconscious performance is at chance-level). The authors apply this logic to a novel analysis technique for inferring conscious performance on the basis of subjective and objective performance thresholds (“surplus coefficient” or SC). Using SC, the authors reveal that the conservative position that unconscious performance is at chance-level cannot hold. Critically, “the proportion of trials for which participants reported being able to identify the content of the visual scene was not sufficient to explain global objective categorization over all trials.”

In general, the authors do a fantastic job of outlining the existing literature in their introduction, the assumptions their study adopts, and the novelty their method brings. The authors employ a tachistoscope to target two types of conscious perception: detection and identification. This procedure allows them to psychophysically distinguish two thresholds of visual awareness with respect to urban and natural scenes. Firstly, the threshold between "not seeing" and "seeing", and secondly, the threshold between "not conscious of content" and "conscious of content". I believe this procedure is a worthwhile contribution to this literature. However, I have reservations about the other major element of their paper—the novel method for inferring unconscious perception. This is why I consider the data to only partly support their conclusions.

Major issue: My concern is the derivation of “surplus coefficient” (SC). The goal of this analysis is to determine the theoretical level of performance on “conscious” trials that would be required to explain “objective performance”, assuming chance-level performance on “unconscious” trials. The upshot of this process is that when SC is greater than 0, it is impossible to explain “objective performance” on the basis of conscious trials alone because “conscious performance” needs to be greater than 100% accurate. This logic seems to hold in the context of this experiment when we consider conscious “detection”. This is because the target of a detection report is the mere perception of a stimulus—we can assume conscious performance will be roughly equivalent for all stimuli on conscious trials. This is not the case for conscious “identification”. In this context, the target of the report is one of 160 urban or 160 natural scenes. It is very likely that this judgment is easier or harder for certain stimuli, even under the assumption that conscious access is a dichotomous process. Given this, it seems mistaken to assume “conscious performance” will be fixed for all conscious trials (and vice versa for “unconscious performance”) when deriving SC (“we verified whether conscious performance could, on its own, explain objective performance”). Despite this concern, I think it’s unlikely stimulus variability entirely discounts the authors’ findings in the “identification” condition. However, it will affect the replicability of this study and extensions were a different set of stimuli used. This issue should be addressed when discussing the results and when using the phrase “direct unconscious perception” in their Discussion - it seems SC, in its present form, is not quite appropriate in the case of conscious identification.

We understand the point developed here and thank the reviewer for this interesting insight. The referee points out that when it comes to identifying the stimuli, it may be that some are harder to categorize than others. We agree with this observation. One way to address this problem would be to use only one image of each category (one urban scene and one natural scene). However, in this case, participants will quickly learn characteristic traits of each image so as to distinguish them from each other, which would preclude us from making any claim about the semantic character of the underlying categorization processes. When the level of categorization is lower, the two subjective thresholds of detection and identification converge towards each other over time, as we observed in a pilot to this experiment. Our intent was to separate these thresholds by using complex images. But we are convinced that variability is necessary to explore unconscious categorization. To reduce image-bound variability, we matched all images in luminance and constrast. However, we recognize the point is important and now acknowledge the limitations raised here in the discussion of the article, line 630.

Stimulus variability is a common issue in the psycholinguistic literature that is accounted for by treating stimuli as a random factor in the error term of a mixed model. This approach may be appropriate here, for instance modelling objective performance as a function of psychometric thresholds, slopes, and conscious and unconscious performance as well as including stimuli and subjects as random factors. Another advantage of this approach is that it allows the inclusion of all trials in the model, and would be robust to changes in the proportion of conscious and unconscious trials + changes in threshold as the experiment progresses.

We thank the reviewer for this suggestion. We have some trouble seeing how to exactly implement a mixed model in the current context. It seems to us that the current set of data cannot be analyzed through a mixed-model approach, for the following reasons:

First, psychometrics’ parameters are fitted for each participant all along the experiment (for subjective detection, subjective identification and objective identification) through a staircase procedure (PSI). As a consequence, we have a parameters’ estimation of only one psychometric per trial. Another consequence of this procedure is that the best estimate is the final estimate, as it is typically calculated through a Bayesian procedure on all previous trials. There is no real sense in using intermediate estimates.

Second, conscious performance is calculated through an equation that uses the parameters of the psychometrics. In addition, unconscious performance is the direct difference between the conscious performance and the global performance. In this context, conscious perception, unconscious perception and the psychometrics parameters are quite collinear, which further hampers the use of mixed models with these parameters as main effects.

In sum, it seems to us that neither the local estimation of the parameters of the psychometrics or unconscious performance are amenable to a mixed model approach.

Minor issues:

As per PLOS ONE guidelines, “contacting author” is not sufficient data availability. I recommend uploading the data to Figshare in .csv format along with a ‘readme’ .txt document that describes each variable.

The results are now available on https://osf.io/s62kv/. 

I was a bit uncertain about the response process and timing in Phase 2, especially for stimulus “present” trials. My understanding is that participants made a “natural” vs. “urban” judgment and a subjective report on every trial (i.e., “absent”, “catch”, and “test” trials). The critical difference was that on “catch” trials the presentation time was determined from responses in Phase 1. But was this the time corresponding to “detection” or “identification” reports?

We used both “subjective detection” and “subjective identification” thresholds. Catch trials were therefore divided into 2 categories: A first category with the duration from subjective detection and the other with duration from subjective identification. We modified this point in the manuscript, line 252.

Line 201 “same stimuli stimulus” —> “same stimuli”

Line 307 “detecting and identify” —> “detecting and identifying”

Line 308 “on the subjective as” —> “on the subjective scale as”

Line 387 “Data are simulating” —> “Simulated data”

Line 399 “same than objective” —> “same as objective”

Line 412 “below than 60% correct” —> “below 60% correct”

Line 421 “was of 0.50” —> “was 0.5”

Line 515 “and of false” —> “and false”

Line 538 “enabled to investigate” —> “enabled us to investigate”

We thank the Reviewer for his careful reading and apologize for the many typos. They have now all been corrected throughout the manuscript.

 

Reviewer #2: This paper describes a psychophysical experiment on unconscious categorization of complex visual images (natural vs. urban scenes) presented with exposure durations in the submillisecond range using a custom built LCD-tachistoscope. Subjective perception (detection/visibility) was evaluated in a first phase (no objective task), followed by a second phase (objective task) where participants had to decide what the stimulus was on each trial (forced choice categorization). In this second phase, subjects had also to perform after each trial the same subjective task as in the first phase (absence of stimulus, stimulus detection or stimulus identification). The authors explored the distribution of responses on the subjective scale for each stimulus exposure duration, through psychometric fitting, and compared it to objective responses. Results indicate that participants are able to correctly decide about the content of a stimulus (that is, to correctly categorize it) even when they fail to identify the contents per se, thus suggesting direct unconscious perception (for this specific setup using complex visual scenes).

This study provides interesting novel results that have implications on the understanding of conscious and subliminal perception when using complex visual stimuli. The paper is well written, and the Methods and Results are thoroughly described.

I think the discussion could be improved by considering other results from other papers that have used a similar approach to render the stimuli subliminal. There is at least one other paper (to my knowledge) that has used a similar device (https://onlinelibrary.wiley.com/doi/10.1002/hbm.22716) with submillisecond exposure duration. In this experiment the authors used rather simple visual stimuli (checkerboards) and found that although the checkerboards were undetectable on a conscious level for the 250 microsecond duration (see their experiment 3), yet they still evoked a brain response (even if a small one only). The authors should relate their present findings to this, at least with respect to the method used.

We did mention the seminal article from Sperdin et al. (2013) in the supplementary material, however we missed this reference. We had this reference in the article, line 194. We looked for other articles by the same author about the usage of a modern tachistoscope, but we did not find further references. In recent correspondence, Herzog mentioned that their tachistoscope had been dismantled. 

 

Reviewer #3: This study presents an experiment in which subjects need to subjectively evaluate the visibility (phase 1) of natural or urban scenes. In a next phase, the need to objectively categorize these scenes and also give a subjective visibility evaluation. The surplus coefficient is introduced as new measure to explain the proportion of objective performance that cannot be explained by the subject’s performance on conscious trials only. The results show that subjects can classify scenes even when they fail to subjectively fail to see the contents of the stimulus. Overall, I found this an interesting study to read with some intriguing results. However, I am not convinced that the use of the surplus coefficient is bringing a lot of novelty or a better measure to the field of consciousness research. I listed my comments below.

- I find I it hard to fully appreciate the surplus coefficient (SC) as the dependent measure. While I get the intuition of the SC, some aspects still remain obscure to me after several readings of the analysis paragraphs. I understand that what is called global performance in Figure 5 is the objective performance, and that the % conscious trials in the figure corresponds to the subjective performance (it would help if there is more consistency in the naming). 

We harmonized the notation of global performance and objective performance throughout the manuscript. The “% conscious trials” in Figure 5 corresponds to the % of stimuli rated as conscious versus unconscious by the participant on the subjective scale, not to subjective performance.

If I understand correctly, the counterintuitive suggestion of an objective performance that needs to be higher than 100% is needed to fit equation (3) and because unconscious performance (UP) is assumed to be equal to 50% in this equation. When looking at the simulations, however, I get the impression that the authors could use a simpler measure than this. If it is assumed that consciousness is an all-or-none phenomenon, the threshold for subjective/objective consciousness can be calculated as the inflexion point of the logistic regression curve (see Del Cul et al., 2007, PLoS Biol). According to my understanding of the surplus coefficient it should be highly similar to the difference in these thresholds: If the objective threshold is lower than the subjective threshold, this suggests that there is some additional – unconscious - processing taking place without reaching the subjective threshold (or a positive SC; Fig. 5D). In the case of an ideal observer (Fig.5C), the objective threshold is equal to the subjective threshold (the difference between the two is 0; or the SC is undetermined), suggesting that there is no additional unconscious processing. In the unlikely case of a negative SC (Fig. 5B), the subjective threshold is lower than the objective threshold which is also highly unlikely as it would mean that subjects can’t objectively classify information that they subjectively label as conscious. The two measurements (difference between thresholds and the SC) seem to behave the same. Because the use of the SC as a new metric is one of the most important claims of this study, it would be nice to understand why the authors constructed this new measure rather than to directly compare the thresholds and why is it beneficial to use the SC. 

We thank the reviewer for this insightful comment. However, we think our approach goes beyond the mere comparison of thresholds by considering that the entire subjective psychometric function is actually important to determine the contribution of conscious and unconscious perception for global performance. We further develop this point in the following.

First, in the introduction, we argue (1) that the threshold of consciousness (i.e., the transition between unconscious and conscious processing) cannot be reduced to a single point of the curve, and (2) that it is dissociable from the threshold of the subjective psychometric function (i.e., the inflexion point) — the paragraph that begins on line 93 is about the necessity to study the entire distribution. In fact, the inflexion point of the subjective psychometric represents the stimulus duration where half of the trials are categorized as conscious and half of the trials are categorized as unconscious. However, the threshold of consciousness — that is, the transition from unconscious to conscious perception — is represented by the entire psychometric function: it corresponds to the probability for the different presentation times to result in a conscious experience or not (e.g., for the 500 µs stimulus presentation time, about 30% of the trials were categorized as having been consciously detected; for the 700 µs presentation time, about 70% were categorized as consciously detected, see Figure 9). Thus, the way in which we define the “threshold of consciousness” constrains how we compare the inflexion points of the objective and subjective psychometrics but, actually, all the other points of the curves as well. In short, we wanted to have the possibility to measure the potential contribution of unconscious perception all along the psychometric curve. 

Second, it is important to clarify that we not only need to compare the thresholds of the objective and subjective psychometrics, but also the slopes of the functions. Figure 5 actually demonstrates that the slope of the psychometric indeed has an impact on the SC and thus on the size of the potential unconscious effect. As an illustration, if you consider the case where the threshold of the subjective psychometric function is lower than the threshold of the objective psychometric function, the SC will be present (superior at 0 on Figure 5) or absent (inferior at 0, blue color) depending on the slope value of the subjective psychometric function. 

The results with the SC appear to be very much in line with more traditional analysis approaches (e.g., aren’t the results presented in Fig. 6 giving you the exact same conclusions about unconscious processing?).

It is true that Figure 6, in which accuracy is computed for the different categories of the subjective scale, might suggest that there is unconscious perception. But this way of computing unconscious perception is prone to a number of potential flaws, as we develop in introduction (see also Balsdon T, Clifford CW 2018), flaws that our method makes it possible to avoid. Indeed, with the approach used in Figure 6, we are simply unable to exclude that the merely stems from a decisional bias: unconscious performance might in fact be driven by a few trials that are mistakenly categorized as “unconscious” (i.e., guess) but that are in fact conscious (i.e., little/almost confident). In this context, one cannot know if the “unconscious effect” comes only from a response bias or from participants’ sensitivity. With the SC method, we are able to control for this bias and show that there is an unconscious effect explained by the differential sensitivity elicited by the objective and subjective tasks.

- In the discussion, the authors write that their results show that unconscious performance is only observable when participants knew there was something on the screen (lines 592-597). This statement appears to be rather paradoxical - a conscious sensation is necessary to have unconscious processing? – and needs to be rewritten. 

We thank the reviewer for this comment. We changed the sentence as follows: “By reductio ad absurdum, we conclude that direct unconscious perception (20) is necessary to explain global performance in our task, and more precisely that objective identification appears to be possible without subjective identification. However, our results fail to show the necessity of unconscious categorization when participants report being unaware that something was presented on the screen (“absent” responses): objective identification is not possible when the participant failed to detect the stimulus — a pattern that is the landmark of the “blindsight effect” (43–48).”

Also, later in the discussion it is written that the current results strongly favor unconscious performance what seems to be contradicting what was written before. The authors need to spell out more in detail what they exactly refer to: Objective identification appears to be possible without subjective identification, but is impossible without subjective detection, or something like this.

We thank the reviewer for the comment. We changed the sentence as follows: “Despite the conservative position and methods we adopted, our results show that performance on subjectively identified trials was not sufficient to explain the objective performance on the objective categorization task, favoring the interpretation that unconscious performance is strongly present when the participant know that something was presented without knowing what it is.”

- What is missing from the experiment is an objective detection task. It would have been informative to see how this threshold relates to the subjective detection threshold and the objective identification threshold.

We agree that this could be an interesting question: is there some unconscious detection performance even when the subject cannot detect the stimulus consciously? We were actually thinking of running a follow-up study on this question. 

Here, we wanted to focus on identification so as to be able to distinguish two levels of non-conscious perception: blindsight (i.e., good objective identification performance in the absence of conscious detection), and unconscious identification (i.e., good objective identification performance in the absence of conscious identification). With an objective detection task, we cannot have this distinction anymore: the unconscious effect would only correspond to a blindsight effect.

- Abstract, line 31: “without being to identify it”. Missing word.

- Page 9, line 201: “…stimulus stimulus…”. Remove stimulus.

- Page 19, line 412: “… who were below than 60%...”. Remove than.

- Page 19, line 420-421: “…mean objective performance was of 0.50…”. Remove of.

- Figure caption of figure 9 says that the yellow curve represents the proportion of correct responses on trials reported as having been subjectively detected. Is this not just the proportion of trials indicated as subjectively seen? Same for the red curve.

We thank the Reviewer for having pointed out these different typos.

- Page 26, line 593:”…our results fail to show the necessity of unconscious categorization…”. Strange sentence.

The sentence was clarified.

- Page 26, line 600: It is concluded that subjective reports were less sensitive in phase 2 what would suggest that performing the objective task biases subjective responses. However, only a change in the slope was observed and no change in the threshold. Does this then reflect a change in bias?

The objective task did not bias the subjective response in the sense that it did not change the criterion from an SDT point of view, which means that it did not make a participant more conservative or liberal. However, and this is a critical point, we observed that the objective task biases the subjective response in the sense that it is reduced compared to Phase 1, when the focus is on the subjective task only. In this case, participants presumably needed to exert increased cognitive effort to maintain information in (working) memory until they produced a subjective report, as compared to phase 1. 

- Page 27, line 623: “…a hypothesis that we infirm.” Should be confirm.

- Page 27, line 626: “…that performance on subjectively identify trials..”. Should be rewritten.

The sentences were rewritten.

---

## [Decision Letter · Decision Letter 1]

9 Jul 2020

Unconscious categorization of sub-millisecond complex images

PONE-D-19-20674R1

Dear Dr. Beauny,

We’re pleased to inform you that your manuscript has been judged scientifically suitable for publication and will be formally accepted for publication once it meets all outstanding technical requirements. 

Within one week, you’ll receive an e-mail detailing the required amendments. When these have been addressed, you’ll receive a formal acceptance letter and your manuscript will be scheduled for publication. At this point, if you could have a look at the few remaining comments of Reviewer 3.

Kind regards,

Magdalena Ewa Król, Ph.D.

Academic Editor

PLOS ONE

Additional Editor Comments (optional):

Reviewers' comments:

Reviewer's Responses to Questions

**Comments to the Author**

1. If the authors have adequately addressed your comments raised in a previous round of review and you feel that this manuscript is now acceptable for publication, you may indicate that here to bypass the “Comments to the Author” section, enter your conflict of interest statement in the “Confidential to Editor” section, and submit your "Accept" recommendation.

Reviewer #1: All comments have been addressed

Reviewer #3: (No Response)

2. Is the manuscript technically sound, and do the data support the conclusions?

Reviewer #1: Yes

Reviewer #3: Yes

3. Has the statistical analysis been performed appropriately and rigorously? 

Reviewer #1: Yes

Reviewer #3: Yes

4. Have the authors made all data underlying the findings in their manuscript fully available?

Reviewer #1: Yes

Reviewer #3: Yes

5. Is the manuscript presented in an intelligible fashion and written in standard English?

Reviewer #1: Yes

Reviewer #3: Yes

6. Review Comments to the Author

Reviewer #1: The authors have revised their manuscript which is now clearer, addresses my comments, and appears to account for comments raised by my fellow reviewers. I also note that their data is now uploaded to the OSF with a comprehensive readme file.

One paper the authors may wish to raise when discussing the impact of image variability on subjective thresholds is Rahnev & Fleming (Neuroscience of Consciousness, 2019). The analysis of Rahnev & Fleming demonstrates that stimulus variability (albeit for simple gabor patches) introduced during staircase procedures can improve metacognitive ability (the correspondence between objective detection reports and subjective confidence ratings). Although there are several major differences in research design, the results of Rahnev & Fleming may lend further support to the conclusions in the present study. Assuming a tight relationship between conscious access and metacognitive ability, stimulus variability (rather than a limitation) may have enhanced subjective thresholds relative to conditions with low stimulus variability. This means the SC-estimates in the present study are possibly more conservative than they might be were stimulus variability controlled more precisely.

In any case, this is a fine addition to the literature on unconscious perception.

Reviewer #3: All of my previous comments have been addressed.

Some minor comments:

I would change stimulus intensity to stimulus duration. Intensity can also be varied by changing brightness, for example, and keeping the duration constant. Because this experiment explicitly manipulates duration, it should be written accordingly.

Line 614: “… a hypothesis that we infirm.” Infirm?

Line 647: ‘…performance on subjectively identified [identify] trials…”

The order of the figures is very confusing in the pdf but the figure numbers do not correspond anymore to the references in the text. This should be checked again. Also in the response to my comment, a reference is made to figure 5 but I guess this should be figure 4 in the revised manuscript.

From line 488 onwards, ‘hits rates’ should be changed to ‘hit rates’.

7. PLOS authors have the option to publish the peer review history of their article (what does this mean?). If published, this will include your full peer review and any attached files.

Reviewer #1: **Yes: **Julian Matthews

Reviewer #3: No

---

## [Editor Report · Acceptance letter]

16 Jul 2020

PONE-D-19-20674R1 

Unconscious categorization of sub-millisecond complex images 

Dear Dr. Beauny:

I'm pleased to inform you that your manuscript has been deemed suitable for publication in PLOS ONE. Congratulations! Your manuscript is now with our production department. 

Kind regards, 

on behalf of

Dr. Magdalena Ewa Król 

Academic Editor

PLOS ONE